# Implicit Regularization of SGD Reduces Shortcut Learning

**Nahal Mirzaie**[1][*]   **Alireza Alipanah**[1][†]   **Ali Abbasi**[1][†]   **Amirmahdi Farzane**[2][†]
**Hossein Jafarinia**[1]   **Erfan Sobhaei**[3]   **Mahdi Ghaznavi**[1]   **Amir Najafi**[1]
**Mahdieh Soleymani Baghshah**[1], and **Mohammad Hossein Rohban**[1]

[1]Computer Engineering Department, Sharif University of Technology
[2]Computer Engineering Department, University of Tehran
[3]Department of Mathematical Sciences, Sharif University of Technology

## Abstract

Training with stochastic gradient descent (SGD) at moderately large learning rates has been observed to improve robustness against spurious correlations, strong correlation between non-predictive features and target labels. Yet, the mechanism underlying this effect remains unclear. In this work, we identify batch size as an additional critical factor and show that robustness gains arise from the implicit regularization of SGD, which intensifies with larger learning rates and smaller batch sizes. This implicit regularization reduces reliance on spurious or shortcut features, thereby enhancing robustness while preserving accuracy. Importantly, this effect appears unique to SGD: gradient descent (GD) does not confer the same benefit and may even exacerbate shortcut reliance. Theoretically, we establish this phenomenon in linear models by leveraging statistical formulations of spurious correlations, proving that SGD systematically suppresses spurious feature dependence. Empirically, we demonstrate that the effect extends to deep neural networks across multiple benchmarks. Our code is available at https://github.com/mirzanahal/sgd-implicit-regularization-shortcuts.

## 1   Introduction

The primary goal of generalization in machine learning is to develop models that perform robustly across diverse realizations of one or more distributions. However, this goal is often compromised when models rely on *shortcuts*, or spurious features: features that are correlated with the target in the training distribution but unstable across environments (Geirhos et al., 2020). Such spurious correlations impede the learning of invariant features that remain stable across different distributions. As a result, models that achieve high accuracy on the training data can fail dramatically on both in-distribution and out-of-distribution samples (Koh et al., 2021b; Puli et al., 2022).

This phenomenon persists even in the presence of Fully Informative Invariant Features (FIIF), which can perfectly predict the label. Despite their predictive power, gradient-based optimizers often select solutions that rely on spurious features (Puli et al., 2023; Nagarajan et al., 2021). In these settings, the label is conditionally independent of the spurious feature given the invariant one, and the Bayes-optimal predictor under the training distribution depends solely on the invariant feature. Nevertheless, models that use both invariant and spurious features typically achieve lower empirical loss than those that rely only on the invariant feature (Arjovsky et al., 2019; Puli et al., 2022; Geirhos et al., 2020). This occurs because spurious features, while less predictive, often increase the *margin* in margin-sensitive loss functions, making solutions that include them more attractive to gradient-based optimization (Soudry et al., 2018). In other words, even when the invariant feature alone suffices for perfect separation, incorporating spurious features can reduce the empirical loss by amplifying the margin.

---

[*]Corresponding to: nahal.mirzaie@ce.sharif.edu
[†]Equal contribution.

The impact of data-dependent factors—such as the strength of spurious correlations and the geometry of the data—on a model's reliance on shortcuts has been extensively studied in linear settings where a FIIF coexists with a spurious feature (Puli et al., 2023; Xue et al., 2024; Nagarajan et al., 2021). In these works, gradient-based optimizers are often treated as black boxes that converge to the max-margin solution, while the role of training hyperparameters, such as batch size $b$ and learning rate $\epsilon$, in modulating shortcut reliance remains poorly understood. Empirically, higher learning rates have been observed to reduce shortcut dependence and improve robustness (Idrissi et al., 2022; Puli et al., 2023; Barsbey et al., 2025), yet this phenomenon cannot be fully explained within existing frameworks. Consequently, the mechanisms by which gradient-based optimizer hyperparameters influence shortcut learning remain unclear, representing an important open question.

## 1.1 Four-Point Data Generating Model

To concretely study the phenomenon described above, researchers have introduced a simple yet theoretically rich data-generating model. A widely used instance is the *four-point model*, defined in two dimensions: a *Fully Informative Invariant Feature* (FIIF) and a spurious feature. Despite its simplicity, this model exposes fundamental limitations of empirical risk minimization (ERM) algorithms, including gradient-descent methods in linear classification, and has become a standard framework for theoretical investigations in out-of-distribution generalization and shortcut learning (Rosenfeld et al., 2021; Puli et al., 2023; Xue et al., 2024; Nagarajan et al., 2021; Ahuja et al., 2021). The model continues to inspire research and raises several open questions.

**Definition 1.1** (Four-Point Data Model). Let $\mathrm{Rad}$ denote the Rademacher distribution over $\{-1, 1\}$. Fix parameters $\rho \in (0, 1)$ and $B > 1$. The data distribution $\mathbb{P} = \mathbb{P}_{\rho, B}(\boldsymbol{X}, y)$ is defined hierarchically as:

$$y \sim \mathrm{Rad}, \quad z \mid y \sim \begin{cases} 1 - \rho & \text{if } z = y, \\ \rho & \text{if } z = -y, \end{cases} \tag{1}$$

and $\boldsymbol{X} \triangleq [\, y, \, Bz \,]$.

Here, $X_1 = y$ is the FIIF, and $X_2 = Bz$ is the spurious feature. The scaling factor $B > 1$ amplifies the influence of $X_2$, so for large $B$, models tend to rely on the spurious feature over the invariant one. A natural, margin-sensitive hypothesis set and cost function in this setting is the class of linear classifiers $\mathcal{H} \triangleq \{\boldsymbol{X} \mapsto \boldsymbol{w}^\top \boldsymbol{X} \mid \boldsymbol{w} \in \mathbb{R}^2, \|\boldsymbol{w}\|_2 \leq 1\}$ with the exponential loss

$$C(\boldsymbol{w}) \triangleq \frac{1}{n} \sum_{i=1}^n e^{-y_i(\boldsymbol{w}^\top \boldsymbol{X}_i)}, \tag{2}$$

where $\{(\boldsymbol{X}_i, y_i)\}_{i=1}^n$ are i.i.d. samples from Definition 1.1. Although $X_1$ can fully predict the label, the spurious feature $X_2 = Bz$ can increase the margin and reduce the loss when $\rho$ is not too large. Consequently, the learned classifier $\boldsymbol{w}^* = [w_y^*, w_z^*]$ depends strongly on the specific optimization algorithm (e.g., GD, SGD), its hyperparameters such as learning rate $\epsilon$ and batch size $b$, as well as the choice of the loss function.

While prior work has mainly studied the effect of the data-generating parameters $B$ and $\rho$ on shortcut reliance (Puli et al., 2023; Nagarajan et al., 2021; Xue et al., 2024; Sagawa et al., 2020b), our focus shifts to training hyperparameters—specifically, batch size $b$ and learning rate $\epsilon$ roles through the lens of the implicit regularization of GD and SGD algorithms.

## 1.2 Our Contribution

We theoretically examine how Gradient Descent (GD) and Stochastic Gradient Descent (SGD) influence reliance on spurious features in the setting of Section 1.1. We then extend these theoretical insights with extensive experiments on real-world datasets, empirically validating the results across a broader family of loss functions and classifiers, including cross-entropy loss and deep neural networks (see Section 4). In summary, we report two striking findings:

- SGD reduces reliance on spurious features, with this effect strengthening as the batch size $b$ decreases and/or the learning rate $\epsilon$ increases. We provide explicit, non-asymptotic guarantees (see Theorem 3.2).

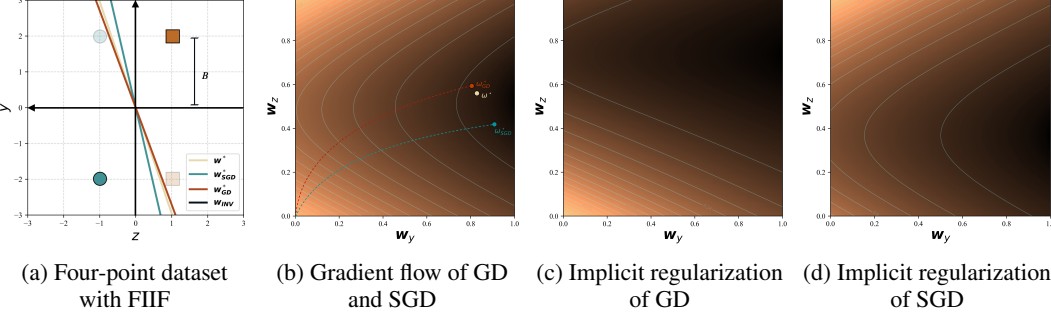

| (a) Four-point dataset with FIIF | (b) Gradient flow of GD and SGD | (c) Implicit regularization of GD | (d) Implicit regularization of SGD |

Figure 1: **Implicit Regularization of GD and SGD on Four-Point dataset with FIIF. (a)** Four-point dataset with a FIIF. The data lie in two dimensions: an invariant feature ($y$) and a spurious feature ($z$). The majority of samples are shown in saturated colors. The invariant solution $\boldsymbol{w}_{\text{INV}}$ achieves perfect classification. Similarly, the solutions that minimize $C(\boldsymbol{w})$, $C_{\text{GD}}(\boldsymbol{w})$, and $C_{\text{SGD}}(\boldsymbol{w})$, denoted by $\boldsymbol{w}^\star$, $\boldsymbol{w}_{\text{GD}}^\star$, and $\boldsymbol{w}_{\text{SGD}}^\star$, respectively, also achieve perfect accuracy, but with larger margin with respect to majority samples. **(b)** Comparison of the minima of $C(\boldsymbol{w})$, $C_{\text{GD}}(\boldsymbol{w})$, and $C_{\text{SGD}}(\boldsymbol{w})$, along with schematic trajectories illustrating the flows of SGD (*blue line*) and GD (*red line*). **(c, d)** Implicit regularization of GD and SGD. Darker regions indicate lower values. Notably, the implicit regularization of SGD imposes a weaker penalty on solutions with smaller $\boldsymbol{w}_z$, thereby favoring parameters that rely less on the spurious feature.

- In contrast, GD does not confer the same benefit and may even slightly increase reliance on shortcuts (see Figure 1). This behavior is also supported theoretically (Theorem 3.1).

A key insight into why the above phenomena occur is that, for non-negligible learning rates $\epsilon > 0$, neither GD nor SGD exactly follows the *gradient flow*, i.e. gradient descent with an *infinitesimal* learning rate, of the original loss $C(\boldsymbol{w})$ (Smith et al., 2021; Barrett & Dherin, 2021). Instead, both methods approximately follow the gradient flow of a *modified* (surrogate) loss that augments the original cost with an additional regularization term, known as the implicit regularization of GD and SGD.

The mechanisms of implicit regularization differ between the GD and SGD. For GD, the regularization penalizes the squared norm of the full-batch gradient, $\|\nabla C(\boldsymbol{w})\|^2$, favoring flatter minima. In addition, SGD also penalizes the average squared norm of the gradients across $m \triangleq \frac{n}{b}$ non-overlapping mini-batches, thereby reducing gradient variance between mini-batches. This difference, both theoretically and empirically, leads to markedly different behaviors in group robustness and reliance on spurious features. Specifically, stronger implicit regularization in SGD, scaling with the learning rate to batch size ratio, more effectively suppresses gradient variance across mini-batches, yielding more consistent performance across subpopulations and greater reliance on invariant features.

Our work aims to provide a foundation for understanding shortcut learning through the lens of optimizer dynamics. In particular, we clarify the seemingly paradoxical benefits of large learning rates and show that combining small batch sizes with appropriately tuned higher learning rates introduces a favorable inductive bias toward robustness against spurious features. This perspective reduces the need for exhaustive hyperparameter searches in explicit shortcut-mitigation methods (Kirichenko et al., 2023; Qiu et al., 2023; Ghaznavi et al., 2025), by leveraging the natural regularization effects of SGD.

## 2 THE CORE IDEA: IMPLICIT REGULARIZATION OF GD AND SGD

In the continuous limit, gradient descent with an infinitesimal step size $\epsilon$ is described by the ordinary differential equation (ODE):

$$\frac{\mathrm{d}}{\mathrm{d}t}\widetilde{\boldsymbol{w}}^{(t)} = -\nabla C(\widetilde{\boldsymbol{w}}^{(t)}), \quad \forall t > 0, \tag{3}$$

which defines the *gradient flow*. GD approximates this continuous process with discrete updates:

$$\boldsymbol{w}^{(t+1)} = \boldsymbol{w}^{(t)} - \epsilon \nabla C(\boldsymbol{w}^{(t)}), \quad \forall t = 0, 1, 2, \ldots. \tag{4}$$

This discretization introduces deviations from the exact gradient flow trajectory. As a result, the optimizer effectively follows the gradient flow of a *modified* loss function that includes an additional regularization term, this is known as **implicit regularization**. It is called "implicit" because no explicit penalty is added; rather, the discretization inherent in GD biases the trajectory away from steep regions with large gradients. Formally, this implicit regularization can be expressed as (Barrett & Dherin, 2021):

$$C_{\mathrm{GD}}(\boldsymbol{w}) \triangleq C(\boldsymbol{w}) + \frac{\epsilon}{4} \big\| \nabla C(\boldsymbol{w}) \big\|^2. \tag{5}$$

Consequently, GD trajectories are repelled not only from regions of high loss but also from areas with large gradient norms. A formal statement is given in Lemma A.3. A similar characterization holds for SGD, where the implicit regularization is expressed in terms of the average squared norm of the mini-batch gradients (Smith et al., 2021):

$$C_{\mathrm{SGD}}(\boldsymbol{w}) \triangleq C(\boldsymbol{w}) + \frac{\epsilon}{4m} \sum_{k=0}^{m-1} \big\| \nabla \widehat{C}_k(\boldsymbol{w}) \big\|^2, \tag{6}$$

with $m$ denoting the number of mini-batches and $\nabla \widehat{C}_k(\boldsymbol{w})$ represents the gradient computed over the $k$-th mini-batch. This result is formalized in Lemma A.4.

We now sketch the key idea behind our theoretical analysis. In the four-point data generation model, the SGD modified loss can be decomposed as:

$$C_{\mathrm{SGD}}(\boldsymbol{w}) = C_{\mathrm{GD}}(\boldsymbol{w}) + \frac{\epsilon \, \mathsf{Var}(\rho_{1:m})}{4} \, f(\boldsymbol{w}; B, \widehat{\rho}), \tag{7}$$

where $\widehat{\rho}$ is the empirical estimate of $\rho$ based on the dataset $\{(\boldsymbol{X}_i, y_i)\}_{i=1}^n$, $\mathsf{Var}(\rho_{1:m})$ denotes the variance of $\rho$ estimates across the $m$ mini-batches of size $b$, and $f(\boldsymbol{w}; B, \widehat{\rho})$ is a function with specific properties. Notably, the magnitude of the second term scales as $\epsilon/b$. We show that $f(\cdot)$ effectively shifts the optimal solution toward smaller $w_z$ (less reliance on the spurious feature) and larger $w_y$ (more reliance on the invariant feature), explaining why SGD with appropriate hyperparameters suppresses shortcut learning.

## 3 MAIN RESULTS

Our main theoretical results are presented in this section. The following theorem shows that under two conditions: (i) sufficiently large $B$, and (ii) sufficiently large $n$ so that the sample mean $\bar{\rho}$ is close to its population mean $\rho$, Gradient Descent (GD) provably increases reliance on spurious features. This theorem provides a asymptotic guarantee, showing that the degree of worsening scales linearly with the learning rate $\epsilon$.

**Theorem 3.1** (Main Result on Gradient Descent). *Assume the four-point data generation model described in Definition 1.1 with parameters $\rho \in (0, \frac{1}{3})$ and $B > 1$. Let $\mathcal{D} = \{(\boldsymbol{X}_i, y_i)\}_{i=1}^n$ be a dataset of $n$ i.i.d. samples drawn from this model. Assume*

$$B \geq \frac{3}{2} \log\left(\frac{1-\rho}{\rho}\right) \quad , \quad n \geq 288 \cdot \log\frac{2}{\zeta},$$

*for some $\zeta \in (0, 1)$. Let $w_{z,\mathrm{GD}}^*$ denote the solution obtained using gradient descent with a sufficiently small step size $\epsilon > 0$ (such that Lemma A.3 applies), and $w_z^*$ be the solution using gradient flow of the original loss. Then, there exists a constant $C > 0$, depending only on $B$ and satisfying $C = \Theta(1)$ with respect to $B$, such that*

$$w_{z,\mathrm{GD}}^* - w_z^* \geq C\epsilon\sqrt{\rho(1-\rho)} + \mathcal{O}(\epsilon^2), \tag{8}$$

*with probability at least $1 - \zeta$ with respect to the randomness of drawing $\mathcal{D}$.*

This shows that GD could result in a higher $w_z^\star$ with respect to the true optimizer, worsening the reliance on the spurious feature. The proof is given in Appendix A.3.

In contrast to GD, SGD has the potential to yield the opposite effect. This benefit, however, only manifests when the minibatch size $b$ is sufficiently small, large $b$ recovers the GD regime, where the improvement disappears. The following theorem formalizes this phenomenon:

**Theorem 3.2** (Main Result on Stochastic Gradient Descent). *Assume the four-point data generation model described in Definition 1.1 with parameters $\rho \in (\frac{1}{100}, \frac{1}{3})$ and $B > 1$. Let $\mathcal{D} = \{(\boldsymbol{X}_i, y_i)\}_{i=1}^{n}$ be a dataset of $n$ i.i.d. samples drawn from this model. Suppose that*

$$B \geq \frac{3}{2} \log\left(\frac{1-\rho}{\rho}\right), \quad n \geq \max\left\{ \frac{8b^3 \log(\frac{2}{\zeta})}{(\rho(1-\rho))^2}, \frac{2b \log(\frac{4}{\zeta})}{\rho(1-\rho)} \right\},$$

*for some $\zeta \in (0,1)$. Let $w_{z,\mathrm{SGD}}^*$ denote the solution obtained using stochastic gradient descent with a sufficiently small step size $\epsilon > 0$ and minibatch size $b \geq 1$ over a single epoch, and let $w_z^*$ be the solution using gradient flow. Then, there exist constants $C_1, C_2 > 0$, depending on $(B, \rho)$ and satisfying $C_1, C_2 = \widetilde{\Theta}(1)$ with respect to both parameters, such that*

$$w_{z,\mathrm{SGD}}^* - w_z^* \leq C_1 \, \epsilon \sqrt{\rho\,(1-\rho)} - \frac{C_2 \, B\epsilon}{b\sqrt{\rho\,(1-\rho)}} + \mathcal{O}(\epsilon^2 + B^{-1}), \tag{9}$$

*with probability at least $1 - \zeta$ with respect to the randomness of $\mathcal{D}$.*

The proof is deferred to Appendix A.3. The key observation is that the negative term dominates whenever $B$ is sufficiently large and/or $b$ is small, which implies $w_{z,\mathrm{SGD}}^* < w_z^*$, that is, SGD shifts the solution toward the invariant feature. Moreover, just as in Theorem 3.1, the effect scales linearly with $\epsilon$, so larger learning rates intensify the phenomenon.

**Corollary 3.3** (Upper bound on minibatch size). *In the setting of Theorem 3.2, stochastic gradient descent provably reduces the reliance on the spurious feature $Bz$ provided that*

$$b \leq \widetilde{\Theta}\left(\frac{B}{\rho(1-\rho)}\right),$$

*for a sufficiently small step size $\epsilon > 0$.*

Intuitively, small mini-batches inject gradient variance that counteracts shortcut reliance. Our analysis shows that whenever $\rho$ is small or $B$ is large, there exists an explicit upper bound on $b$ below which SGD provably improves robustness. In other words, stronger spurious correlations, arising from smaller $\rho$ or larger $B$, require smaller batch sizes to guarantee that SGD effectively suppresses shortcut learning. The dependence on $\epsilon$ is natural, while the required sample size $n$ remains moderate rather than prohibitively large.

The core contribution of this work is the rigorous analysis of the four-point data-generating model under the exponential loss and linear classifiers. To further clarify why the same mechanism should persist beyond this stylized setting, we also provide a general result under mild assumptions. In particular, in the regime $n, m \to \infty$, suppose there exists a shortcut solution $\boldsymbol{w}_{\mathrm{bad}}$ that minimizes the empirical risk, and a group-robust solution $\boldsymbol{w}_{\mathrm{good}}$ that is not a minimizer. If, for every minibatch, the gradient discrepancy between majority and minority subpopulations is uniformly smaller at $\boldsymbol{w}_{\mathrm{good}}$ than at $\boldsymbol{w}_{\mathrm{bad}}$, then SGD's modified loss with implicit regularization assigns a smaller penalty to $\boldsymbol{w}_{\mathrm{good}}$, thereby favoring it relative to the shortcut solution $\boldsymbol{w}_{\mathrm{bad}}$. In contrast, full-batch GD lacks this variance-dependent effect and instead amplifies the preference for the $\boldsymbol{w}_{\mathrm{bad}}$. A formal statement and proof are provided in Appendix A.4.

## 4 EXPERIMENTS

We extend our analysis beyond theory with empirical evaluations on deep models (MLP, ResNets He et al. (2015), and Bert Devlin et al. (2019b)) trained on established spurious-correlation benchmarks. We focus on regimes achieving near-optimal in-distribution generalization, as indicated by stable accuracy (ACC). We report worst-group accuracy (WGA) as a metric of shortcut reliance, where higher WGA reflects stronger core feature learning and reduced dependence on spurious correlations. These results confirm that strengthening the implicit regularization of SGD effectively improve WGA.

### 4.1 NON-MONOTONIC EFFECT OF LEARNING RATE ON GROUP ROBUSTNESS

Theoretically, the strength of implicit regularization of SGD scales with increasing the learning rate Smith et al. (2021). In practice, however, very small learning rates lead to under-training, while

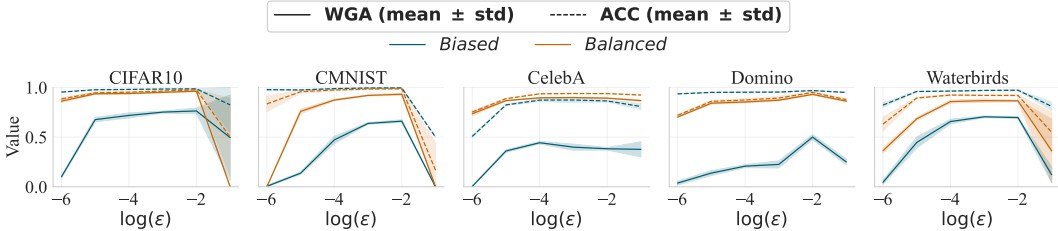

Figure 2: **Effect of Learning Rate on WGA and ACC.** Test set ACC and WGA are reported for a fixed batch size of $b = 128$ on both balanced and biased datasets with spurious correlation ($\rho = 5\%$). As shown, once ACC reaches an optimal or near-optimal level, WGA continues to increase with learning rate up to an optimal point, beyond which both ACC and WGA decline.

Table 1: **Best WGA across batch sizes** ($b$). On datasets with spurious correlation ($\rho = 5\%$), we report for each batch size the highest mean WGA achieved across six learning rates. Within each dataset, the maximum value is highlighted in blue, the minimum in orange, and $\Delta$ denotes the difference between them.

| | CMNIST | Domino | Waterbirds | CelebA | CIFAR10 |
|---|---|---|---|---|---|
| $b$ | $WGA$ | $WGA$ | $WGA$ | $WGA$ | $WGA$ |
| 8 | $67.5_{\pm 1.1}$ | $59.3_{\pm 5.3}$ | $79.7_{\pm 0.8}$ | $46.0_{\pm 1.0}$ | $80.1_{\pm 2.2}$ |
| 16 | $68.4_{\pm 1.2}$ | $56.3_{\pm 3.6}$ | $77.7_{\pm 2.8}$ | $51.9_{\pm 6.8}$ | $78.9_{\pm 1.7}$ |
| 32 | $68.0_{\pm 1.9}$ | $49.4_{\pm 5.1}$ | $73.2_{\pm 8.7}$ | $45.3_{\pm 1.8}$ | $79.6_{\pm 1.7}$ |
| 64 | $67.5_{\pm 0.6}$ | $51.9_{\pm 3.6}$ | $67.9_{\pm 5.0}$ | $40.5_{\pm 2.4}$ | $78.8_{\pm 2.8}$ |
| 128 | $66.0_{\pm 1.3}$ | $50.0_{\pm 2.6}$ | $70.4_{\pm 0.9}$ | $44.3_{\pm 1.3}$ | $76.3_{\pm 2.8}$ |
| 256 | $64.7_{\pm 0.9}$ | $43.4_{\pm 2.1}$ | $68.2_{\pm 1.9}$ | $45.0_{\pm 1.3}$ | $77.9_{\pm 2.5}$ |
| $\Delta$ | $+3.7$ | $+15.9$ | $+11.5$ | $+11.4$ | $+3.8$ |

excessively large learning rates destabilize training or cause divergence Goodfellow et al. (2016); Nocedal & Wright (2006); Smith et al. (2021).

Within the regimes that an near-optimal in-distribution-generalization is achievable, and for a fixed batch size, increasing the learning rate, equivalently, increasing $\log(\epsilon)$, improves the WGA, as reflected in the solid curves (Figure 2). This effect is especially pronounced in biased datasets relative to balanced ones, as illustrated by the blue and orange curves, respectively. Overall, the results reveal a non-monotonic relationship between the learning rate and group robustness: WGA increases with the learning rate, reaches a near-optimal point, and subsequently declines, likely due to convergence instability. Once the learning rate becomes sufficiently large to achieve near-optimal in-distribution generalization, further increases in learning rate result in improved robustness and yield higher WGA. However, when the learning rate grows too large, it hinders convergence, and therefore both ACC and WGA decline. This pattern is consistent across different batch sizes and datasets (see Figure 6).

## 4.2 SMALLER BATCH SIZES, STRONGER GROUP ROBUSTNESS

After confirming the effect of learning rate on group robustness, we investigate whether reducing batch size enhances group robustness. Figure 3 presents both ACC and WGA.in biased datasets with spurious correlation ($\rho = 5\%$), where the hatched bars correspond to ACC and the solid bars to WGA. Although datasets exhibited varying sensitivities of WGA to batch size, a consistent trend emerged: once in-distribution generalization is saturated, training with smaller batch sizes improved the final WGA of the model. These results suggest that, once the learning rate secures strong generalization, reducing the batch size also guides the optimizer toward solutions that are more robust across both minority and majority groups. This effect is also observed in Transformer architectures for language datasets(Table 2).

Furthermore, we optimized the learning rate separately for each batch size to determine the maximum WGA achievable under that configuration. Across all datasets, the highest WGA occurred

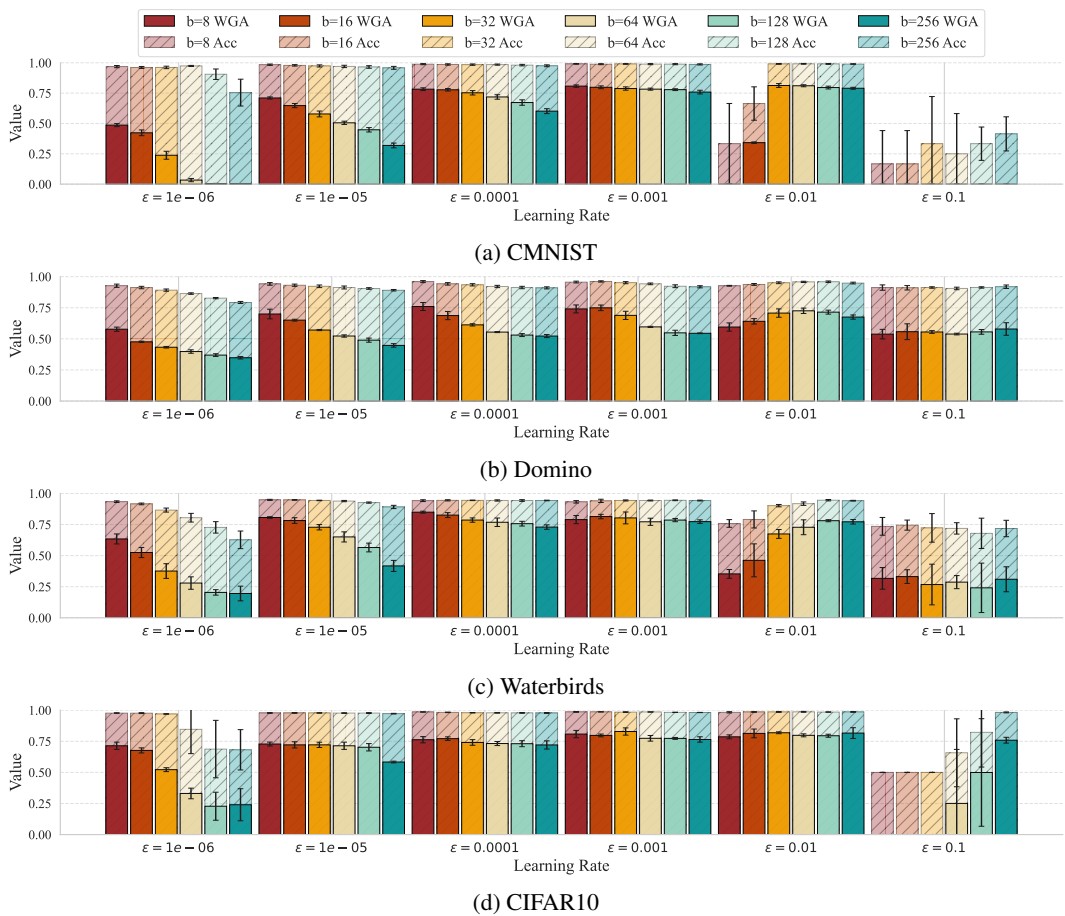

Figure 3: **Joint Effect of Learning Rate and Batch Size on WGA and ACC.** When in-distribution generalization is ensured, smaller batch sizes consistently yield higher WGA across all datasets, indicating improved robustness. For extremely high or low learning rates, in-distribution generalization fails; in these regimes, reliable conclusions about robustness cannot be drawn.

Table 2: **WGA and ACC across batch sizes on language datasets.** WGA and ACC for Multi-NLI and CivilComments across batch sizes for a fixed learning rate.

| | | **Multi-NLI** | | | **CivilComments** | |
|---|---|---|---|---|---|---|
| $b$ | $\varepsilon$ | WGA | ACC | $\varepsilon$ | WGA | ACC |
| 8 | $10^{-4}$ | $76.75_{\pm 0.48}$ | $82.18_{\pm 0.04}$ | $10^{-5}$ | $60.72_{\pm 3.30}$ | $91.76_{\pm 0.26}$ |
| 16 | $10^{-4}$ | $76.58_{\pm 2.04}$ | $82.40_{\pm 0.07}$ | $10^{-5}$ | $59.73_{\pm 1.68}$ | $92.12_{\pm 0.12}$ |
| 32 | $10^{-4}$ | $76.50_{\pm 0.57}$ | $81.74_{\pm 0.20}$ | $10^{-5}$ | $54.89_{\pm 4.39}$ | $92.34_{\pm 0.17}$ |
| 64 | $10^{-4}$ | $75.80_{\pm 1.85}$ | $81.93_{\pm 0.13}$ | $10^{-5}$ | $53.40_{\pm 0.34}$ | $92.30_{\pm 0.02}$ |
| 128 | $10^{-4}$ | $75.78_{\pm 0.68}$ | $80.96_{\pm 0.05}$ | $10^{-5}$ | $53.70_{\pm 0.74}$ | $92.02_{\pm 0.02}$ |
| 256 | $10^{-4}$ | $75.17_{\pm 0.63}$ | $79.34_{\pm 0.02}$ | – | – | – |
| $\Delta$ | | **+1.58** | | | **+7.32** | |

consistently with smaller batch sizes (8 or 16), while the lowest values were typically observed at the largest batch sizes (64 or higher) (see Table 1). These results indicate that smaller batch sizes systematically enhance group robustness and allow the model to reach fundamentally higher ceilings of group robustness.

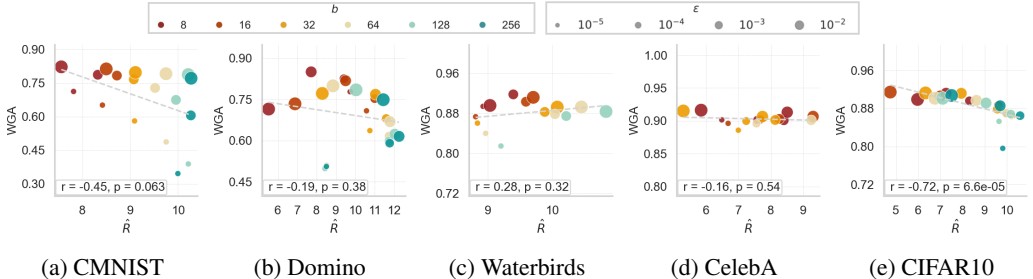

(a) CMNIST  (b) Domino  (c) Waterbirds  (d) CelebA  (e) CIFAR10

Figure 4: **Scatter Plot and Pearson Correlation between $\hat{R}$ and WGA.** A negative correlation between $\hat{R}$ and WGA indicates that models converging to minima with more optimized implicit regularization terms, exhibiting greater robustness to spurious correlations.

### 4.3 IMPLICIT REGULARIZATION FOR GROUP ROBUSTNESS

A key question is whether the observed improvement in group robustness under stronger implicit regularization arises from a distinct mechanism that enhances robustness, or if it is merely a consequence of overall improvements in standard generalization. To address this, we monitor improvements in ACC and WGA. WGA is substantially more sensitive to variations in batch size and learning rate than ACC (Table 5, and Figure 3). Once the learning rate is within a range that ensures in-distribution generalization, fluctuations in WGA are therefore much more pronounced than those observed in ACC (Table 5). These findings indicate that improvements in group robustness are not simply a byproduct of uniform gains in generalization. Instead, they reflect a distinct mechanism by which the choice of batch size and learning rate directs the model toward solutions that have better performance on minority groups.

To examine the relationship between implicit regularization and group robustness more rigorously, we select models based on best validation performance and compute both the implicit regularization term of SGD and worst-group accuracy (WGA) on a biased test set that matches the training distribution. Recall that the implicit regularization of SGD can be computed as: $R = \frac{1}{m}\sum_{i=1}^{m}\|\nabla\widehat{C}_i\|^2$. For consistency across experimental settings, we normalize $R$ by $b$ and report its logarithm, denoted by $\widehat{R}$ (see Figure 4). As we expected the more robust solutions exhibit lower gradient variance across mini-batches and consequently lower $\widehat{R}$.

To intuitively clarify why penalizing gradient variance across mini-batches improves group robustness, we view each mini-batch as a sampled "domain" from the training distribution. Smaller batch sizes increase variability in subpopulation composition across mini-batches, resulting in fluctuating spurious correlations. Suppressing gradient variance under these shifts promotes reliance on stable, invariant features rather than shortcuts, thereby improving robustness. A very similar principle that has been observed in domain generalization Rame et al. (2022); Shi et al. (2022). Therefore, stronger suppression of gradient variance across mini-batches corresponds to improved robustness and reduced shortcut reliance.

### 4.4 SMALL BATCH SIZE AS A TRICK FOR MULTI-LEVEL SPURIOUS CORRELATIONS

Existing methods for mitigating shortcut learning perform well on single-level spurious datasets such as CelebA, Waterbirds, and CMNIST. However, on more realistic datasets that contain complex or unknown spurious attributes, such as multi-level spurious correlations in Domino-CMF, these methods often perform at or below random chance on the minority groups Ghaznavi et al. (2025). This happens because, even after applying these methods, the second-level spurious attribute remains encoded in the learned representation and cannot be corrected when the algorithm only targets the first (known) spurious correlation (see Figure 5). In these scenarios, training with small batch sizes provides a stronger inductive bias for current explicit methods. As shown in Table 2, applying DFR, AFR, and EVaLS with small batch sizes yields substantial WGA improvements of 25–37% compared to training the same methods with larger batch sizes.

| Method | Group Info | Waterbirds | | CelebA | | CMNIST | | Domino-CMF | |
|---|---|---|---|---|---|---|---|---|---|
| | Train/Val | Small | Large | Small | Large | Small | Large | Small | Large |
| DFR | ✓/✓ | $90.6_{\pm0.7}$ | $\mathbf{92.9_{\pm0.2}}$ | $86.5_{\pm2.3}$ | $\mathbf{88.3_{\pm1.1}}$ | $\mathbf{84.98_{\pm0.3}}$ | $79.73_{\pm0.9}$ | $72.9_{\pm4.7}$ | $42.7_{\pm2.7}$ |
| AFR | ✗/✓ | $77.8_{\pm6.6}$ | $90.4_{\pm1.1}$ | $71.5_{\pm2.8}$ | $82.0_{\pm0.5}$ | $72.92_{\pm1.7}$ | $68.11_{\pm1.1}$ | $67.3_{\pm5.6}$ | $40.3_{\pm0.5}$ |
| EVaLS | ✗/✗ | $78.4_{\pm0.6}$ | $88.4_{\pm3.1}$ | $56.0_{\pm10.5}$ | $85.3_{\pm0.4}$ | $78.3_{\pm1.9}$ | $73.37_{\pm6.2}$ | $\mathbf{76.7_{\pm1.7}}$ | $51.2_{\pm1.4}$ |
| ERM | ✗/✗ | $74.5_{\pm2.9}$ | $67.4_{\pm0.6}$ | $59.9_{\pm5.1}$ | $47.7_{\pm3.3}$ | $71.3_{\pm1.2}$ | $65.2_{\pm0.8}$ | $59.0_{\pm4.3}$ | $36.8_{\pm2.0}$ |

Table 3: **Effect of batch size on explicit methods performance.** Results are grouped by dataset with comparisons between small and large batch sizes. The best result for each dataset is bolded, and colored cells highlight the higher WGA between small and large batch settings for each method. Notably, on the multi-level spurious dataset Domino-CMF, using small batch sizes with explicit methods yields particularly significant gains.

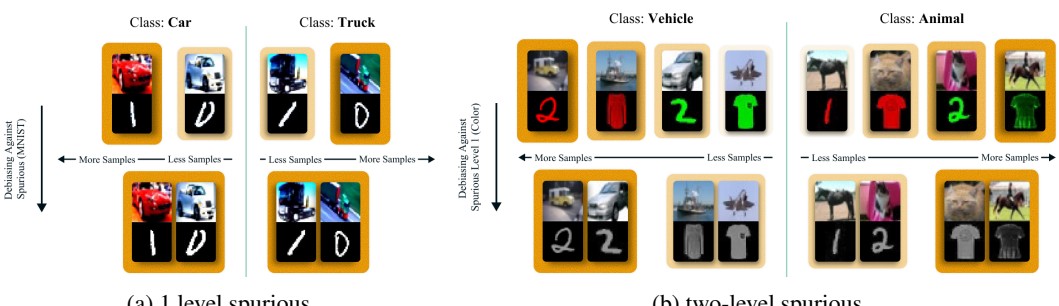

(a) 1 level spurious        (b) two-level spurious

Figure 5: **Illustration of datasets with 1 and 2 level of spurious(a)** Domino dataset Murali et al. (2023) pairs a label-bearing CIFAR-10 image with a same-class Fashion-MNIST tile that adds a spurious shortcut. Debiasing against MNIST shortcut result balanced groups. **(b)** Domino-CMF Ghaznavi et al. (2025) uses a CIFAR-10 top and a red/green MNIST or FashionMNIST bottom, introducing two level of spurious correlations (style and color). Debiasing against color, lead to still a biased dataset based on FashionMNIST shape.

## 5 RELATED WORK

### 5.1 BATCH SIZE AND LEARNING RATE FOR IN-DISTRIBUTION GENERALIZATION

Prior work on in-distribution generalization consistently shows that small-batch SGD outperforms large-batch training. Studies focusing on batch size find that, for any fixed learning rate, generalization peaks at a moderate (non-large) batch size, while very large batches degrade performance (Keskar et al., 2017; Smith & Le, 2018). Most studies examine batch size and learning rate jointly, demonstrating that their ratio controls solution sharpness and, consequently, generalization (Goyal et al., 2018; Jastrzebski et al., 2017; Chaudhari & Soatto, 2018; Park et al., 2019). Yet, how these mechanisms extend beyond in-distribution generalization remains underexplored.

### 5.2 BATCH SIZE AND LEARNING RATE FOR ROBUSTNESS

The effects of batch size and learning rate on robustness are more nuanced. In adversarial training, Wang et al. (2024); Yao et al. (2018) vary the inner mini-batch size used to generate adversarial examples and show that increasing this batch size improves certified robustness up to a moderate scale, after which gains saturate. Beyond adversarial settings, smaller batches have been shown to improve performance under data imbalance (Shwartz-Ziv et al., 2023). In contrast, robustness to label noise favors the opposite regime, large batch sizes and small learning rates, highlighting a tension between hyperparameter choices that address different robustness objectives (Rolnick et al., 2018). More recently, higher learning rates have been observed to reduce shortcut reliance and improve robustness (Idrissi et al., 2022; Puli et al., 2023; Barsbey et al., 2025). Despite these empirical findings, the mechanisms linking learning rate and especially batch size, to group robustness remain largely unexplained.

### 5.3 Connection to Domain and Out-of-Distribution Generalization

In the domain generalization literature, reducing gradient variance across domains has been consistently shown to improve out-of-distribution performance Rame et al. (2022); Shi et al. (2022). In this work we show that a similar principle arises in the context of mini-batch training: each mini-batch can be interpreted as a domain sampled from the overall training distribution. Smaller batch sizes increase the likelihood that a mini-batch will contain a disproportionate representation of underrepresented samples relative to the majority, thereby inducing varying spurious correlations. Controlling the variance of mini-batch gradients effectively enforces robustness to such distributional shifts. In other words, minimizing gradient variance across mini-batches encourages the model to rely on features that remain stable across changing correlation patterns, thereby promoting out-of-distribution robustness.

## 6 Discussion

Our study highlights how learning rate and batch size influence reliance on spurious features, through the lens of implicit regularization. We show that SGD mitigates shortcut dependence, especially with smaller batch sizes and larger learning rates, while GD offers no such benefit and can even exacerbate it. The distinction arises from different mechanism of implicit regularization of these algorithms: GD penalizes full-batch gradient norms leading to flatter minima, whereas SGD also reduces gradient variance across mini-batches, fostering robustness for underrepresented groups. This explains the advantages of high learning rates and small batches for group robustness, showing that robustness can emerge naturally from SGD's inductive bias. We hope this work lays a foundation for further research and reduces extensive hyperparameter tuning by encouraging smaller batch sizes to leverage SGD's implicit regularization.

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

## A   THEORETICAL ANALYSIS

### A.1   PRELIMINARIES

This section introduces our notation and reviews a set of relevant theorems and theoretical results from prior work. We also present our main statistical data model: the four-point data generation process.

#### A.1.1   NOTATION

Let $[n]$ denote $\{1, 2, \ldots, n\}$ for $n \in \mathbb{N}$. Vectors are denoted by bold letters, e.g. $\boldsymbol{X}$, and scalars are written in ordinary letters, for example $y$. For a vector $\boldsymbol{X}$ and for $i \leq j$, by $\boldsymbol{X}_{i:j}$ we mean the subvector of $\boldsymbol{X}$ from the $i$th component to the $j$th component.

Assume a binary classification problem with feature vector $\boldsymbol{X} \in \mathcal{X}$ and binary label $y \in \{-1, 1\}$. Here, $\mathcal{X}$ could be any measurable space, but we usually consider $\mathcal{X} \subseteq \mathbb{R}^d$ for some dimension $d \in \mathbb{N}$. In our analysis, we consider a linear classifier of the form

$$\mathcal{F} \triangleq \left\{ f_{\boldsymbol{w}} : \boldsymbol{X} \mapsto \boldsymbol{w}^\top \boldsymbol{X} \,\middle|\, \boldsymbol{w} \in \mathbb{R}^d, \, \|\boldsymbol{w}\|_2 \leq 1 \right\},$$

where $\mathcal{F}$ represents the set of linear classifiers without bias. For a given feature-label pair $(\boldsymbol{X}, y)$ and a classifier $f_{\boldsymbol{w}}$, we define the exponential loss $\ell(y, f_{\boldsymbol{w}}(\boldsymbol{X}))$ as

$$\ell(y, f_{\boldsymbol{w}}(\boldsymbol{X})) = e^{-y f_{\boldsymbol{w}}(\boldsymbol{X})}. \tag{10}$$

This loss, when averaged, provides a smooth, margin-sensitive estimate of the classification error rate.

### A.1.2 DATA GENERATION MODEL

We adopt a recurrent statistical data model used in related works, particularly in the literature on spurious features, known as the *four-point data model*. This simple construction captures fundamental effects of spurious features, while several theoretical and practical aspects of this model are still unresolved.

**Definition A.1** (Four-Point Data Generation Process with FIIF). Assume $d \geq 2$. Let $\mathrm{Rad}$ denote the Rademacher distribution over $\{-1, 1\}$, and let $\mathcal{N}$ denote the zero-mean Gaussian distribution with identity covariance $\boldsymbol{I}$. Fix parameters $\rho \in (0, 1)$ and $B > 1$. The data distribution $\mathbb{P} = \mathbb{P}_\rho(\boldsymbol{X}, y)$ is defined via the following hierarchical process:

$$y \sim \mathrm{Rad},$$
$$z|y \sim \begin{cases} 1 - \rho & \text{if } z = y, \\ \rho & \text{if } z = -y, \end{cases}$$
$$\boldsymbol{X}_{3:d} \sim \mathcal{N}(\boldsymbol{0}, \boldsymbol{I}_{d-2}). \tag{11}$$

We then define $\boldsymbol{X} \triangleq [\, y, \, Bz, \, \boldsymbol{X}_{3:d} \,]$. In this construction, $X_1 = Bz$ is the spurious feature, $X_2 = y$ is the FIIF, and the remaining $d - 2$ coordinates are independent Gaussian noise.

For some $n \in \mathbb{N}$, assume the dataset $\mathcal{D} \triangleq \{(\boldsymbol{X}_i, y_i) | i \in [n]\}$ consists of $n$ i.i.d. samples from $\mathbb{P}_\rho$ for some fixed but unknown $\rho \in (0, 1)$ and $B > 1$. Also, the empirical average loss with respect to $\mathcal{D}$ and for any $\boldsymbol{w} \in \mathbb{R}^d$ with $\|\boldsymbol{w}\|_2 \leq 1$ is defined as

$$C(\boldsymbol{w}) \triangleq \frac{1}{n} \sum_{i=1}^{n} \ell(y_i, f_{\boldsymbol{w}}(\boldsymbol{X}_i)). \tag{12}$$

**The Good vs. Bad Solutions.** In the above model, the first component of the feature vector, $X_1 = y$, always provides a perfect estimate of the label. The spurious component, $X_2 = Bz$, coincides with the label with probability $1 - \rho$, in which case it yields a larger margin for $B > 1$. However, with probability $\rho$, this component provides an incorrect estimate of the label. The remaining $d-2$ components of $\boldsymbol{X}$ are independent of the label and therefore irrelevant for prediction. We are interested in those solutions $\boldsymbol{w}$ of the linear model that assign significant weight to $X_1$ while placing small weights on the remaining components.

For now, assume $d = 2$, i.e., only the core and the spurious components are present. In this case, we partition the parameter space of $\boldsymbol{w} = [w_y, w_z]^\top \in \mathbb{R}^2$ into *good* and *bad* solutions according to the model's relative reliance on the spurious feature $z$ versus the core feature $y$.

**Definition A.2** (Good and Bad Solutions). For any $\boldsymbol{w} = [w_y, w_z]^\top \in \mathbb{R}^2$, we call $\boldsymbol{w}$ *bad* if $Bw_z > w_y$, and *good* otherwise, i.e., $Bw_z \leq w_y$.

In words, a solution is classified as *bad* if the weighted contribution of the spurious feature (scaled by $B$) dominates that of the core feature; otherwise, it is classified as *good*.

### A.1.3 BACKGROUND THEORY

Barrett and Dherin Barrett & Dherin (2021) analyzed the effect of finite learning rates on the dynamics of gradient descent (GD) using *backward error analysis*, a technique from numerical ODE theory Hairer et al. (2006). Recall that the gradient descent method applied to a differentiable loss $C(\boldsymbol{w})$ with step size $\epsilon > 0$ is given by

$$\boldsymbol{w}^{(t+1)} \leftarrow \boldsymbol{w}^{(t)} - \epsilon \nabla C(\boldsymbol{w}^{(t)}), \quad \forall t = 0, 1, 2, \ldots.$$

A continuous-time analogue of GD is the so-called *gradient flow* (GF), formulated as the ODE

$$\frac{\mathrm{d}\,\widetilde{\boldsymbol{w}}^{(t)}}{\mathrm{d}t} = -\nabla C(\widetilde{\boldsymbol{w}}^{(t)}), \quad \forall t > 0,$$

which, unlike GD, is defined for all real times $t > 0$. The goal of such analyses is to compare the behavior of the discrete trajectory $\boldsymbol{w}^{(t)}$ with that of the continuous trajectory $\widetilde{\boldsymbol{w}}(t)$ across different cost functions and step sizes $\epsilon$. The key observation of Barrett & Dherin (2021) is that the discrete iterates of GD do not exactly follow the gradient flow of the original cost $C(\boldsymbol{w})$. Instead, they remain close to the trajectory of a *modified* cost function. Formally:

**Lemma A.3** (Trajectory of Gradient Descent, from Barrett & Dherin (2021)). *Consider gradient descent on a differentiable cost $C(\boldsymbol{w})$ with step size $\epsilon > 0$, and assume $\epsilon$ is sufficiently small. Then, given some mild conditions on $C$, the trajectory of the iterates $\boldsymbol{w}^{(t)}$ resembles the gradient flow trajectory, but with respect to the* modified *cost*

$$C_{\mathrm{GD}}(\boldsymbol{w}) \triangleq C(\boldsymbol{w}) + \frac{\epsilon}{4}\big\|\nabla C(\boldsymbol{w})\big\|^2, \tag{13}$$

*where $\nabla C(\boldsymbol{w})$ denotes the gradient of the original cost.*

This result follows from backward error analysis, which shows that for small but finite $\epsilon$, GD iterates track the gradient flow of $C_{\mathrm{GD}}$ rather than of the original $C$.

Smith et al. (2021) introduced an alternative form of backward error analysis that explicitly accounts for correlations between mini-batches within a single training epoch. Using this approach, it is shown that for sufficiently small learning rates, the mean Stochastic Gradient Descent (SGD) iterate after one epoch, averaged over all possible mini-batch orderings, remains close to the trajectory of gradient flow on a *modified cost function*.

**Lemma A.4** (Trajectory of Stochastic Gradient Descent, from (Smith et al., 2021)). *Consider the setting of Lemma A.3, and assume SGD is applied with a sufficiently small step size $\epsilon > 0$, using $m$ non-overlapping mini-batches with $m \ll n$. Then the trajectory of SGD iterates resembles that of a gradient flow, but with respect to the modified cost*

$$C_{\mathrm{SGD}}(\boldsymbol{w}) \triangleq C(\boldsymbol{w}) + \frac{\epsilon}{4m} \sum_{k=0}^{m-1} \left\|\nabla \widehat{C}_k(\boldsymbol{w})\right\|^2, \tag{14}$$

*where $\nabla \widehat{C}_k(\boldsymbol{w})$ denotes the gradient of the empirical loss on the $k$-th mini-batch.*

Analogous to the GD case, this modified loss consists of the original full-batch loss plus an implicit regularization term. However, the structure of this regularizer differs from that of GD, potentially leading to distinct local and global minima.

We note that Lemmas A.3 and A.4 are derived for unconstrained optimization. Under the exponential loss, however, the unconstrained minimizer lies at infinity for separable data, meaning that the norm of the weights grows without bound. To obtain a well-defined solution, we therefore restrict the parameter space to the unit ball $\mathcal{W} \in \{\boldsymbol{w} \in \mathbb{R}^d : \|\boldsymbol{w}\|_2 \leq 1\}$. In this constrained problem, the minimizer necessarily lies on the boundary $\|\boldsymbol{w}\|_2 = 1$.

After the trajectory reaches the boundary, projection removes the radial component of the gradient (which would otherwise increase the norm), and only the tangential component determines how the direction of $\boldsymbol{w}$ evolves along the sphere.

In our analysis, we assume that the first-order modified-loss characterization continues to describe this tangential component of the projected GD/SGD dynamics. Thus, implicit regularization influences not only the loss landscape but also the directional evolution of the classifier within the constrained domain.

## A.2 BASIC LEMMAS

In this section, we present a collection of fundamental lemmas that will be used in the proofs of Section A.3, where we state our main results on the comparison between GD and SGD with gradient flow.

**Lemma A.5** (*Exponential Loss*). *For constants $B > 1$ and $\bar{\rho} \in (0, 1/2)$, and assuming $B \geq \frac{3}{4}\log((1-\bar{\rho})/\bar{\rho})$, consider the constrained program*

$$\min_{w_y, w_z} \quad e^{-w_y}\left(\bar{\rho}\,e^{Bw_z} + (1-\bar{\rho})\,e^{-Bw_z}\right) \quad \text{subject to} \quad w_y^2 + w_z^2 \leq 1. \tag{15}$$

*Then the optimal solution is given by*

$$\left\{ w_z^* = \Gamma\,(1-\Delta),\; w_y^* = \sqrt{1-w_z^{*2}},\; \lambda^* = \frac{e^{-w_y^*}}{2w_y^*}\sqrt{\bar{\rho}(1-\bar{\rho})}\left(e^{B\Gamma\Delta} + e^{-B\Gamma\Delta}\right) \right\}, \tag{16}$$

where $\lambda^\star$ is the Lagrange multiplier at the optimal point and

$$\Gamma \triangleq \frac{1}{2B} \log\left(\frac{1-\bar\rho}{\bar\rho}\right) \quad , \quad \Delta \triangleq \frac{1}{1 + B^2\sqrt{1-\Gamma^2} + \frac{\Gamma}{1-\Gamma^2}} + \mathcal{O}(1/B^3).$$

*Proof.* For notational convenience set $g(w_z) = \bar\rho e^{Bw_z} + (1-\bar\rho)e^{-Bw_z}$, $f(w_y, w_z) = e^{-w_y}g(w_z)$. The optimization problem is

$$\min_{(w_y,w_z)\in\mathbb{R}^2} f(w_y, w_z) \qquad \text{subject to} \qquad w_y^2 + w_z^2 \le 1.$$

Form the Lagrangian with multiplier $\lambda \ge 0$: $\mathcal{L}(w_y, w_z, \lambda) = e^{-w_y}g(w_z) + \lambda(w_y^2 + w_z^2 - 1)$. The KKT conditions consist of

$$\begin{align}
&\text{(i)} \ \ \nabla_w \mathcal{L}(w, \lambda) = 0, \notag \\
&\text{(ii)} \ \ w_y^2 + w_z^2 \le 1, \notag \\
&\text{(iii)} \ \ \lambda(w_y^2 + w_z^2 - 1) = 0, \notag \\
&\text{(iv)} \ \ \lambda \ge 0. \tag{17}
\end{align}$$

Differentiating we obtain

$$\partial_{w_y}\mathcal{L} = -e^{-w_y}g(w_z) + 2\lambda w_y = 0 \ , \ \partial_{w_z}\mathcal{L} = e^{-w_y}g'(w_z) + 2\lambda w_z = 0,$$

where $g'(w_z) = B\big(\bar\rho e^{Bw_z} - (1-\bar\rho)e^{-Bw_z}\big)$, $g''(w_z) = B^2 g(w_z)$. If $\lambda = 0$ then $-e^{-w_y}g(w_z) = 0$, impossible since the left side is strictly negative. Hence $\lambda > 0$, and by complementary slackness $w_y^2 + w_z^2 = 1$. With $\lambda > 0$ the stationarity equations read

$$2\lambda w_y = e^{-w_y}g(w_z), \quad 2\lambda w_z = -e^{-w_y}g'(w_z).$$

Eliminating $\lambda$ gives $\frac{w_z}{w_y} = -\frac{g'(w_z)}{g(w_z)}$. Since $w_y > 0$ (otherwise the first equation would give a contradiction), write $w_y = \sqrt{1-w_z^2}$. Thus the necessary condition is

$$\frac{w_z}{\sqrt{1-w_z^2}} = -\frac{g'(w_z)}{g(w_z)}. \tag{$\star$}$$

Define $\Phi(w_z) = \frac{w_z}{\sqrt{1-w_z^2}}$, $\Psi(w_z) = -\frac{g'(w_z)}{g(w_z)}$. For $\Phi$ we have $\Phi'(w_z) = (1-w_z^2)^{-3/2} > 0$, hence $\Phi$ is strictly increasing. For $\Psi$, $\Psi'(w_z) = -\frac{g''(w_z)g(w_z) - g'(w_z)^2}{g(w_z)^2}$. Substituting $g''(w_z) = B^2 g(w_z)$ and simplifying yields

$$g''(w_z)g(w_z) - g'(w_z)^2 = 4\bar\rho(1-\bar\rho)B^2 > 0,$$

so $\Psi'(t) < 0$. Thus $\Phi$ is strictly increasing, $\Psi$ strictly decreasing; therefore $(\star)$ admits at most one solution. Since the feasible set is compact and $f$ continuous, a minimizer exists, hence there is exactly one solution $w_z^\star \in (-1, 1)$. The minimizer is therefore

$$w_y^\star = \sqrt{1-(w_z^\star)^2}, \quad \lambda^\star = \frac{e^{-w_y^\star}g(w_z^\star)}{2w_y^\star} > 0.$$

Looking back at $(\star)$, it can be equivalently written as

$$\begin{align}
\frac{w_z}{\sqrt{1-w_z^2}} &= -B\frac{\bar\rho\, e^{Bw_z} - (1-\bar\rho)\, e^{-Bw_z}}{\bar\rho\, e^{Bw_z} + (1-\bar\rho)\, e^{-Bw_z}} \notag \\
&= -B\tanh\left(Bw_z + \tfrac{1}{2}\log\frac{\bar\rho}{1-\bar\rho}\right). \tag{18}
\end{align}$$

We now try to approximate $\Delta$ up to an error of at most $)(c^{-2})$. Without loss of generality, let us write $w_z^\star = \Gamma + \Lambda$ with $\Gamma \triangleq \frac{1}{2B}\log(\frac{1-\bar\rho}{\bar\rho})$. Then, we have

$$-B\tanh(B\Lambda) = \frac{\Gamma + \Lambda}{\sqrt{1-(\Gamma+\Lambda)^2}}.$$

By assuming $B\Lambda \ll 1$, we aim to approximate $\Lambda$. Therefore, we have $\tanh B\Lambda \simeq B\Lambda$ and as a result, we have

$$-B^2\Lambda = \frac{\Gamma}{\sqrt{1-\Gamma^2}} + \Lambda\left[\frac{1}{\sqrt{1-\Gamma^2}} + \frac{\Gamma}{(1-\Gamma^2)^{3/2}}\right] + \mathcal{O}(\Lambda^2),$$

which implies the following approximation:

$$\Lambda = -\frac{\Gamma(1-\Gamma^2)}{B^2(1-\Gamma^2)^{3/2} + 1 + \Gamma - \Gamma^2} + \mathcal{O}(\Gamma/B^3)$$

$$\Rightarrow w_z^* = \Gamma(1 - \underbrace{\frac{1}{1 + B^2\sqrt{1-\Gamma^2} + \frac{\Gamma}{1-\Gamma^2}}}_{\Delta}) + \mathcal{O}(\Gamma B^{-3}). \tag{19}$$

Next, we derive an explicit formulation for Lagrange multiplier $\lambda^\star$. We have

$$e^{B\Gamma} = \exp\left(\tfrac{1}{2}\log\frac{1-\bar\rho}{\bar\rho}\right) = \sqrt{\frac{1-\bar\rho}{\bar\rho}} \ , \ e^{-B\Gamma} = \sqrt{\frac{\bar\rho}{1-\bar\rho}}.$$

Hence,

$$g(w_z^*) = \bar\rho\, e^{Bw_z^*} + (1-\bar\rho)\, e^{-Bw_z^*}$$

$$= \bar\rho\sqrt{\frac{1-\bar\rho}{\bar\rho}}e^{B\Gamma\Delta} + (1-\bar\rho)\sqrt{\frac{\bar\rho}{1-\bar\rho}}e^{-B\Gamma\Delta}$$

$$= \sqrt{\bar\rho(1-\bar\rho)}\left(e^{B\Gamma\Delta} + e^{-B\Gamma\Delta}\right). \tag{20}$$

Substituting into $\lambda^\star = \frac{e^{-w_y^*}g(w_z^*)}{2w_y^*}$ completes the proof. $\qquad\square$

**Lemma A.6.** *Let $C : \mathbb{R}^d \to \mathbb{R}$ be a twice continuously differentiable cost function, and assume that its unconstrained minimizer lies strictly outside the unit ball, i.e., $\left\|\arg\min_{\boldsymbol{w}} C(\boldsymbol{w})\right\|_2 > 1$. Consider the constrained optimization problem*

$$(\boldsymbol{w}^*, \lambda^*) \triangleq \arg\min_{\boldsymbol{w}} C(\boldsymbol{w}) \quad \text{subject to} \quad \|\boldsymbol{w}\|_2^2 \le 1, \tag{21}$$

*where, with a slight abuse of notation, $\lambda^*$ denotes the Lagrange multiplier associated with the quadratic constraint. By standard KKT arguments, we have $\lambda^* > 0$ at the optimum. Let $f : \mathbb{R}^d \to \mathbb{R}$ be another differentiable function. For a sufficiently small $\epsilon > 0$, consider the perturbed problem*

$$\widehat{\boldsymbol{w}}^* \triangleq \arg\min_{\boldsymbol{w}} C(\boldsymbol{w}) + \tfrac{\epsilon}{4}f(\boldsymbol{w}) \quad \text{subject to} \quad \|\boldsymbol{w}\|_2^2 \le 1. \tag{22}$$

*Then the perturbed optimizer satisfies the first-order expansion*

$$\widehat{\boldsymbol{w}}^* = \boldsymbol{w}^* - \frac{\epsilon}{8\lambda^*}\left(\boldsymbol{I} - \boldsymbol{w}^*\boldsymbol{w}^{*\top}\right)\left(\boldsymbol{I} + \tfrac{1}{2\lambda^*}\nabla^2 C(\boldsymbol{w}^*)\right)^{-1}\nabla f(\boldsymbol{w}^*) + \mathcal{O}(\epsilon^2). \tag{23}$$

*Proof.* Let $\widehat{\boldsymbol{w}}^* = \boldsymbol{w}^* + \boldsymbol{\delta}$ and denote the perturbed multiplier as $\widehat\lambda^* = \lambda^* + \Delta$. As $\epsilon \to 0$, both $\boldsymbol{\delta}$ and $\Delta$ vanish. The KKT conditions for the perturbed program are

(i) $\quad \nabla C(\boldsymbol{w}^* + \boldsymbol{\delta}) + \tfrac{\epsilon}{4}\nabla f(\boldsymbol{w}^* + \boldsymbol{\delta}) + 2(\lambda^* + \Delta)(\boldsymbol{w}^* + \boldsymbol{\delta}) = 0,$

(ii) $\quad (\lambda^* + \Delta)\left(\|\boldsymbol{w}^* + \boldsymbol{\delta}\|_2^2 - 1\right) = 0,$

(iii) $\quad \lambda^* + \Delta \ge 0. \tag{24}$

Since $\lambda^* > 0$, condition (iii) is automatically satisfied for small $\epsilon$. Expanding (i)–(ii) to first order yields

$(\star) \quad \nabla^2 C(\boldsymbol{w}^*)\,\boldsymbol{\delta} + \tfrac{\epsilon}{4}\nabla f(\boldsymbol{w}^*) + 2\lambda^*\boldsymbol{\delta} + \Delta\boldsymbol{w}^* = \mathcal{O}(\epsilon^2),$

$(\star\star) \quad \boldsymbol{\delta}^\top\boldsymbol{w}^* = \mathcal{O}(\epsilon^2). \tag{25}$

From $(\star)$ we obtain

$$\boldsymbol{\delta} = -\frac{\epsilon}{8\lambda^*}\left(\boldsymbol{I} + \tfrac{1}{2\lambda^*}\nabla^2 C(\boldsymbol{w}^*)\right)^{-1}\nabla f(\boldsymbol{w}^*) + \gamma\boldsymbol{w}^*,$$

where $\gamma$ depends on $\Delta$. Imposing the orthogonality condition ($\star\star$) determines $\gamma$ uniquely, ensuring that $\boldsymbol{\delta}$ has no component in the direction of $\boldsymbol{w}^*$. This yields

$$\boldsymbol{\delta} = -\frac{\epsilon}{8\lambda^*}\left(\boldsymbol{I} - \boldsymbol{w}^*\boldsymbol{w}^{*\top}\right)\left(\boldsymbol{I} + \tfrac{1}{2\lambda^*}\nabla^2 C(\boldsymbol{w}^*)\right)^{-1}\nabla f(\boldsymbol{w}^*) + \mathcal{O}(\epsilon^2),$$

which proves the claim. $\qquad\square$

**Lemma A.7.** *Let $X_1, \ldots, X_n$ be i.i.d.* Bernoulli($\rho$) *random variables with parameter $\rho \in (0, 1/3)$. Fix integers $m \geq 1$ and $b \geq 1$ such that $n = mb$. Partition the indices $\{1, \ldots, n\}$ into $m$ disjoint blocks of size $b$ and let $\widehat{\rho}_i = \frac{1}{b}\sum_{j \in \text{block } i} X_j$, $i = 1, \ldots, m$ be the block (mini-batch) averages. Define the sample variance of the block means by $S^2 \triangleq \frac{1}{m}\sum_{i=1}^m \left(\widehat{\rho}_i - \bar{\rho}\right)^2$, $\bar{\rho} \triangleq \frac{1}{m}\sum_{i=1}^m \widehat{\rho}_i$. Set $\sigma^2 \triangleq \text{Var}(\widehat{\rho}_i) = \frac{\rho(1-\rho)}{b}$. Fix a confidence parameter $\zeta \in (0, 1)$. If*

$$n \geq \max\left\{\frac{8b^3\log(\frac{2}{\zeta})}{(\rho(1-\rho))^2}, \frac{2b\log(\frac{4}{\zeta})}{\rho(1-\rho)}\right\}, \tag{26}$$

*then with probability at least $1 - \zeta$ we have*

$$S^2 \geq \frac{\rho(1-\rho)}{2b} = \frac{\sigma^2}{2}.$$

*Proof.* For each block $i$ write $\widehat{\rho}_i = \frac{1}{b}\sum_{j=1}^b X_{i,j}$ where, for fixed $i$, the $X_{i,1}, \ldots, X_{i,b}$ are i.i.d. Bernoulli($\rho$). Then $\mathbb{E}[\widehat{\rho}_i] = \rho$ and $\text{Var}(\widehat{\rho}_i) = \frac{\rho(1-\rho)}{b} \triangleq \sigma^2$. Introduce the population second moment about the true mean

$$V_{\text{pop}} \triangleq \frac{1}{m}\sum_{i=1}^m (\widehat{\rho}_i - \rho)^2.$$

Using the identity $\frac{1}{m}\sum_{i=1}^m(\widehat{\rho}_i - \bar{\rho})^2 = V_{\text{pop}} - (\bar{\rho} - \rho)^2$, it suffices to ensure simultaneously

$$\text{(A)} \quad V_{\text{pop}} \geq \tfrac{3}{4}\sigma^2 \quad \text{and} \quad \text{(B)} \quad (\bar{\rho} - \rho)^2 \leq \tfrac{1}{4}\sigma^2,$$

for then $S^2 = V_{\text{pop}} - (\bar{\rho} - \rho)^2 \geq \tfrac{3}{4}\sigma^2 - \tfrac{1}{4}\sigma^2 = \tfrac{1}{2}\sigma^2$. We bound the failure probabilities of (A) and (B) using Hoeffding's inequality.

**(A) bound.** Define $Y_i \triangleq (\widehat{\rho}_i - \rho)^2$. The $Y_i$ are i.i.d., satisfy $0 \leq Y_i \leq 1$, and $\mathbb{E}[Y_i] = \sigma^2$. By Hoeffding's inequality, for any $\varepsilon > 0$,

$$\mathbb{P}\left(\frac{1}{m}\sum_{i=1}^m Y_i \leq \sigma^2 - \varepsilon\right) \leq \exp\left(-2m\varepsilon^2\right).$$

Taking $\varepsilon = \sigma^2/4$ yields $\mathbb{P}\left(V_{\text{pop}} < \tfrac{3}{4}\sigma^2\right) \leq \exp\left(-2m(\sigma^2/4)^2\right) = \exp\left(\frac{-m\sigma^4}{8}\right)$.

**(B) bound.** The overall average $\bar{\rho}$ is the average of all $n = mb$ Bernoulli draws. By Hoeffding's inequality, for any $t > 0$,

$$\mathbb{P}\left(|\bar{\rho} - \rho| \geq t\right) \leq 2\exp\left(-2nt^2\right).$$

Set $t = \sigma/2$. Then $\mathbb{P}\left((\bar{\rho} - \rho)^2 > \tfrac{1}{4}\sigma^2\right) \leq 2\exp\left(-2n(\sigma/2)^2\right) = 2\exp\left(-\frac{n\sigma^2}{2}\right)$. Using $n = mb$ and $\sigma^2 = \rho(1-\rho)/b$ we have $n\sigma^2 = m\rho(1-\rho)$, hence

$$\mathbb{P}\left((\text{B) fails}\right) \leq 2\exp\left(-\frac{m\rho(1-\rho)}{2}\right).$$

By the union bound,

$$\mathbb{P}\left((\text{A) fails or (B) fails}\right) \leq \exp\left(-\frac{m\sigma^4}{8}\right) + 2\exp\left(-\frac{m\rho(1-\rho)}{2}\right).$$

To guarantee this is at most $\zeta$ it suffices to require each summand to be $\leq \zeta/2$, i.e. $\exp\left(\frac{-m\sigma^4}{8}\right) \leq \frac{\zeta}{2}$ and $2\exp\left(\frac{-m\rho(1-\rho)}{2}\right) \leq \frac{\zeta}{2}$. Taking logarithms and rearranging gives the sufficient conditions

$$m \geq \frac{8\log(2/\zeta)}{\sigma^4} \quad \text{and} \quad m \geq \frac{2\log(4/\zeta)}{\rho(1-\rho)}.$$

Substituting $\sigma^4 = (\rho(1-\rho))^2/b^2$ yields the condition displayed in equation 26. Under that condition the total failure probability is at most $\zeta$, so with probability at least $1 - \zeta$ both (A) and (B) hold and therefore

$$S^2 \geq \frac{\sigma^2}{2} = \frac{\rho(1-\rho)}{2b},$$

as claimed. The proof is complete. $\qquad\square$

### A.3 EXPONENTIAL RESULTS

Recall the four-point data generation model and the respective exponential loss function from Section A.1. According to this setting, the cost function $C(\boldsymbol{w})$ can be rewritten as follows:

$$
\begin{aligned}
C(\boldsymbol{w}) &= \frac{1}{n} \sum_{i=1}^{n} \ell(y_i, h_{\boldsymbol{w}}(\boldsymbol{X}_i)) \\
&= \frac{1}{n} \sum_{i=1}^{n} e^{-y_i \boldsymbol{w}^\top \boldsymbol{X}_i} \\
&= \frac{1}{n} \sum_{i=1}^{n} e^{-w_y} \left( e^{Bw_z} \mathbb{1}\{z_i \neq y_i\} + e^{-Bw_z} \mathbb{1}\{z_i = y_i\} \right) \\
&= e^{-w_y} \left[ e^{Bw_z} \left( \frac{1}{n} \sum_{i=1}^{n} \mathbb{1}\{z_i \neq y_i\} \right) + e^{-Bw_z} \left( \frac{1}{n} \sum_{i=1}^{n} \mathbb{1}\{z_i = y_i\} \right) \right] \\
&= e^{-w_y} \left( \bar{\rho} \, e^{Bw_z} + (1 - \bar{\rho}) \, e^{-Bw_z} \right),
\end{aligned}
\tag{27}
$$

where

$$
\bar{\rho} \triangleq \frac{1}{n} \sum_{i=1}^{n} \mathbb{1}\{z_i \neq y_i\}
$$

denotes the empirical fraction of samples with misaligned spurious component $z$ with respect to the core label $y$. Note that $\mathbb{E}[\bar{\rho}] = \rho$. Moreover, by a Chernoff bound, for all $\varepsilon > 0$,

$$
\mathbb{P}(\, |\bar{\rho} - \rho| \geq \varepsilon) \leq 2 e^{-\frac{n\varepsilon^2}{2}}.
\tag{28}
$$

In the following subsections, we analyze how i) Gradient Descent (GD) and ii) Stochastic Gradient Descent (SGD) with strictly positive step sizes $\epsilon > 0$ (and, for SGD, a specified number of minibatches $m$) affect convergence to the optimal point of the above loss. In both cases, we compare the resulting solution to the optimal $\boldsymbol{w}$ of the linear classifier at the minimum loss.

Specifically, Theorem A.8 shows that GD, relative to gradient flow, increases the reliance on the spurious feature $z$ by raising the optimal value of $w_z$, which is undesirable.

In contrast, Theorem A.9 surprisingly demonstrates that SGD can reduce reliance on the spurious feature, particularly when small batch sizes are used. Our results hold with high probability over the randomness of the training data and provide strict, concrete, non-asymptotic bounds on the increase or decrease of $w_z$.

#### A.3.1 GRADIENT DESCENT

This theorem is our main result on gradient descent.

**Theorem A.8** (Main Result on Gradient Descent for Exponential Loss). *Assume the four-point data generation model described in Section A.1.2 with parameters $\rho \in (0, \frac{1}{3})$ and $B > 1$. Let $\mathcal{D} = \{(\boldsymbol{X}_i, y_i)\}_{i=1}^{n}$ be a dataset of $n$ i.i.d. samples drawn from this model. Assume*

$$
B \geq \frac{3}{2} \log\left( \frac{1 - \rho}{\rho} \right) \quad , \quad n \geq 288 \cdot \log \frac{2}{\zeta},
$$

*for some $\zeta \in (0, 1)$. Let $w_{z,\mathrm{GD}}^*$ denote the solution obtained using gradient descent with a sufficiently small step size $\epsilon > 0$, and $w_z^*$ be the solution using gradient flow. Then, there exists a constant $C > 0$, depending only on $B$ and satisfying $C = \Theta(1)$ with respect to $B$, such that*

$$
w_{z,\mathrm{GD}}^* - w_z^* \geq C\epsilon \sqrt{\rho\,(1 - \rho)} + \mathcal{O}(\epsilon^2),
\tag{29}
$$

*with probability at least $1 - \zeta$ with respect to the randomness of drawing $\mathcal{D}$.*

The proof is provided after a brief discussion. The theorem considers a four-point data generation model with hardness parameter $B > 1$ (see Section A.1.2) and misalignment probability $\rho$. It states that, as long as $B$ is sufficiently larger than the logarithmic ratio $\log\left(\frac{1-\rho}{\rho}\right)$, and $n$ is moderately

large so that $\bar{\rho}$ and $\rho$ do not deviate significantly, GD always increases reliance on spurious features by increasing $w_z$. The magnitude of this increase, up to first order in the step size $\epsilon > 0$, is proportional to both $\epsilon$ and $\sqrt{\bar{\rho}(1 - \bar{\rho})}$, but does not grow unboundedly with $B$.

The conditions in the theorem are intuitive: when $B$ is very close to 1 (especially for a moderate $\rho$), the spurious feature is far less strong than the core feature from a *margin* perspective. In this regime, reliance on the spurious feature is inherently small, and the analysis becomes cumbersome. Apart from this, no additional restrictions are imposed.

The proof relies on tools from constrained optimization and KKT theory, combined with concentration bounds and basic linear and nonlinear algebra.

*Proof of Theorem A.8.* The proof proceeds in four stages:

- **Concentration of $\bar{\rho}$:** First, we show that the empirical quantity $\bar{\rho}$—the fraction of samples in $\mathcal{D}$ with counter-aligned core and spurious features $(y, z)$—concentrates around the true value $\rho$ with high probability.

- **Gradient of the perturbation term:** We leverage Lemma A.6 and compute the gradient of the additional perturbation term, $\nabla f(\boldsymbol{w}^*)$, where

$$\frac{\epsilon}{4} f(\boldsymbol{w}) \triangleq \frac{\epsilon}{4} \left\| \nabla C(\boldsymbol{w}) \right\|_2^2, \quad \forall \boldsymbol{w},$$

  and $\boldsymbol{w}^*$ denotes the optimal solution of the unperturbed problem (i.e., the gradient flow solution). Lemma A.5 provides an explicit expression for this unperturbed solution $\boldsymbol{w}^*$.

- **Computation of the matrix term:** Next, we compute the second component required by Lemma A.6, namely

$$\left( \boldsymbol{I} - \boldsymbol{w}^* \boldsymbol{w}^{*\top} \right) \left( \boldsymbol{I} + \tfrac{1}{2\lambda^*} \nabla^2 C(\boldsymbol{w}^*) \right)^{-1}.$$

- **Analysis of the final solution:** Finally, we analyze the resulting solution and determine the condition under which it is positive—i.e., when gradient descent increases the reliance on the spurious feature $z$ by enlarging $w_z$. We show that this holds under the stated constraints of the theorem, and we further simplify the magnitude of this increase.

**Concentration of $\bar{\rho}$ around $\rho$:** For the remainder of the proof, we work with the empirical quantity $\bar{\rho}$ in place of the statistical parameter $\rho$, where

$$\bar{\rho} \triangleq \frac{1}{n} \sum_{i=1}^n \mathbb{1}\{z_i \neq y_i\}, \tag{30}$$

since only $\bar{\rho}$ appears in our subsequent formulas. By the Chernoff bound, we have

$$\mathbb{P}(\, |\bar{\rho} - \rho| \geq \varepsilon) \;\leq\; 2\exp\!\left(-\tfrac{n\varepsilon^2}{2}\right). \tag{31}$$

Note that $\rho \leq \frac{1}{3}$, and the condition $B \geq \frac{3}{2} \log\!\left(\frac{1-\rho}{\rho}\right)$ implies $B \geq \frac{3}{4} \log\!\left(\frac{1-\bar{\rho}}{\bar{\rho}}\right)$, provided that $\bar{\rho} \leq \frac{1}{3} + \frac{1}{12}$. Thus, it suffices to ensure $|\rho - \bar{\rho}| < \frac{1}{12}$. Applying the Chernoff bound, we see that this event holds with probability at least $1 - \zeta$ whenever

$$n \;\geq\; 288 \log\!\left(\tfrac{2}{\zeta}\right).$$

Henceforth, we condition on this event and proceed under the assumption that $B \geq \frac{3}{4} \log\!\left(\frac{1-\bar{\rho}}{\bar{\rho}}\right)$.

**Computing $\nabla f(\boldsymbol{w}^*)$:** Next, we aim to compute $\nabla f(\boldsymbol{w})$ at the optimal point $\boldsymbol{w}^*$, where $f$ (as defined above) is the perturbation term which is added to the original cost due to applying Gradient Descent (GD). In this regard, we have the following relations:

$$C(\boldsymbol{w}) = \left( \bar{\rho} e^{Bw_z} + (1 - \bar{\rho}) e^{-Bw_z} \right) e^{-w_y}$$

$$\Rightarrow \quad \nabla C(\boldsymbol{w}) = \left[ -C(\boldsymbol{w}) , \; B e^{-w_y} \left( \bar{\rho} e^{Bw_z} - (1 - \bar{\rho}) e^{-Bw_z} \right) \right]^\top. \tag{32}$$

Therefore, we have

$$
\begin{aligned}
\|\nabla C(\boldsymbol{w})\|_2^2 =& C^2(w) + B^2 e^{-2w_y} \left( \bar{\rho}^2 e^{2Bw_z} + (1-\bar{\rho})^2 e^{-2Bw_z} - 2\rho_i(1-\bar{\rho}) \right) \\
=& (1+B^2) \left( \bar{\rho}^2 e^{2Bw_z} + (1-\bar{\rho})^2 e^{-2Bw_z} \right) e^{-2w_y} \\
& - (B^2 - 1) 2\bar{\rho}(1-\rho_i) e^{-2w_y}.
\end{aligned}
\tag{33}
$$

The above can be represented in the following compact form:

$$
f(w_y, w_z) = e^{-2w_y} \left\{ (1+B^2)\left[ e^{2Bw_z} \bar{\rho}^2 + e^{-2Bw_z}(1-\bar{\rho})^2 \right] - 2(B^2-1)\bar{\rho}(1-\bar{\rho}) \right\}.
$$

We aim to compute the term $\nabla f(\boldsymbol{w}^*)$ at the optimal point of the constrained ordinary loss, which is already carried out in Lemma A.5, i.e.,

$$
\boldsymbol{w}^* = \begin{bmatrix} w_y \\ w_z \end{bmatrix} = \begin{bmatrix} \sqrt{1-w_z^2} \\ w_z \end{bmatrix}, \quad w_z = \Gamma(1-\Delta),
$$

where $\Gamma \triangleq \frac{1}{2B}\log\left(\frac{1-\bar{\rho}}{\bar{\rho}}\right)$ and $\Delta$ is defined according to the lemma. Define the auxiliary function $g(\cdot)$ as

$$
g(w_z) = (1+B^2)\left[ e^{2Bw_z} \bar{\rho}^2 + e^{-2Bw_z}(1-\bar{\rho})^2 \right] - 2(B^2-1)\bar{\rho}(1-\bar{\rho}).
$$

Then $f(w_y, w_z) = e^{-2w_y} g(w_z)$, so the gradient is

$$
\nabla f = \begin{bmatrix} \frac{\partial f}{\partial w_y} \\ \frac{\partial f}{\partial w_z} \end{bmatrix} = \begin{bmatrix} -2e^{-2w_y} g(w_z) \\ e^{-2w_y} g'(w_z) \end{bmatrix}.
$$

This way, we have

$$
\begin{aligned}
g'(w_z) &= (1+B^2)\left[ 2Be^{2Bw_z}\bar{\rho}^2 - 2Be^{-2Bw_z}(1-\bar{\rho})^2 \right] \\
&= 2B(1+B^2)\left[ e^{2Bw_z}\bar{\rho}^2 - e^{-2Bw_z}(1-\bar{\rho})^2 \right],
\end{aligned}
\tag{34}
$$

where the terms can be simplified as follows:

$$
w_z = \frac{1}{2B}\log\frac{1-\bar{\rho}}{\bar{\rho}} + \Gamma\Delta \implies e^{2Bw_z} = \frac{1-\bar{\rho}}{\bar{\rho}}e^{-2B\Gamma\Delta}, \; e^{-2Bw_z} = \frac{\bar{\rho}}{1-\bar{\rho}}e^{2B\Gamma\Delta}.
$$

Then

$$
\begin{aligned}
g'(w_z) &= 2B(1+B^2)[\bar{\rho}(1-\bar{\rho})e^{-2B\Gamma\Delta} - \bar{\rho}(1-\bar{\rho})e^{2B\Gamma\Delta}] \\
&= -4B(1+B^2)\bar{\rho}(1-\bar{\rho})\sinh(2B\Gamma\Delta).
\end{aligned}
\tag{35}
$$

On the other hand, using the same logic, we simply have

$$
g(w_z) = 2(1+B^2)\bar{\rho}(1-\bar{\rho})\left( \cosh(2B\Gamma\Delta) - \frac{B^2-1}{B^2+1} \right).
$$

Hence,

$$
\nabla f(\boldsymbol{w}^*) = -4(1+B^2)\bar{\rho}(1-\bar{\rho})e^{-2w_y} \begin{bmatrix} \cosh(2B\Gamma\Delta) - \frac{B^2-1}{B^2+1} \\ B\sinh(2B\Gamma\Delta) \end{bmatrix}.
$$

**Computing the matrix** $\left(\boldsymbol{I} - \boldsymbol{w}^*\boldsymbol{w}^{*\top}\right)\left(\boldsymbol{I} + \frac{1}{2\lambda^*}\nabla^2 C(\boldsymbol{w}^*)\right)^{-1}$**:** We now aim to find a closed-form formula for the $2 \times 2$ matrix which should be multiplied to $\nabla f(\boldsymbol{w}^*)$. Recall the auxiliary function $g(w_z) = \bar{\rho}e^{Bw_z} + (1-\bar{\rho})e^{-Bw_z}$. Then, $C(\boldsymbol{w}^*) = e^{-w_y}g(w_z)$ and therefore the Hessian matrix can be written as

$$
\nabla^2 C(\boldsymbol{w}^*) = \begin{pmatrix} C & -\partial_{w_z}C \\ -\partial_{w_z}C & B^2 C \end{pmatrix},
$$

with $\partial_{w_z}C = e^{-w_y}B\left(\bar{\rho}e^{Bw_z} - (1-\bar{\rho})e^{-Bw_z}\right)$. Using the special choice of $\Gamma$, we simplify

$$
g(w_z) = 2\sqrt{\bar{\rho}(1-\bar{\rho})}\cosh(B\Gamma\Delta), \quad g'(w_z) = -2\sqrt{\bar{\rho}(1-\bar{\rho})}\sinh(B\Gamma\Delta),
$$

hence

$$
\begin{aligned}
C &= 2e^{-w_y}\sqrt{\bar{\rho}(1-\bar{\rho})}\cosh(B\Gamma\Delta), \\
\partial_{w_z}C &= -2Be^{-w_y}\sqrt{\bar{\rho}(1-\bar{\rho})}\sinh(B\Gamma\Delta).
\end{aligned}
\tag{36}
$$

Since $\lambda^* = \frac{e^{-w_y}}{w_y} \sqrt{\bar{\rho}(1-\bar{\rho})} \cosh(B\Gamma\Delta)$, we obtain

$$\nabla^2 C(\boldsymbol{w}^*) = 2w_y\lambda^* \begin{pmatrix} 1 & B\tanh(B\Gamma\Delta) \\ B\tanh(B\Gamma\Delta) & B^2 \end{pmatrix},$$

which leads to the following formulations:

$$M \triangleq \boldsymbol{I} + \frac{1}{2\lambda^*}\nabla^2 C(\boldsymbol{w}^*) = \begin{pmatrix} 1+w_y & Bw_y\tanh(B\Gamma\Delta) \\ Bw_y\tanh(B\Gamma\Delta) & 1+B^2 w_y \end{pmatrix}.$$

$$M^{-1} = \frac{1}{\det(M)} \begin{pmatrix} 1+B^2 w_y & -Bw_y\tanh(B\Gamma\Delta) \\ -Bw_y\tanh(B\Gamma\Delta) & 1+w_y \end{pmatrix}. \tag{37}$$

where the determinant can be computed as $\det(M) = (1+w_y)(1+B^2 w_y) - B^2 w_y^2 \tanh^2(B\Gamma\Delta)$. Since $\boldsymbol{w} = (w_y, w_z)^\top$ satisfies $w_y^2 + w_z^2 = 1$, the tangent-space projection operator can be written as

$$P \triangleq \boldsymbol{I} - \boldsymbol{w}^*\boldsymbol{w}^{*\top} = \begin{pmatrix} 1-w_y^2 & -w_y w_z \\ -w_y w_z & 1-w_z^2 \end{pmatrix} = \begin{pmatrix} w_z^2 & -w_y w_z \\ -w_y w_z & w_y^2 \end{pmatrix}.$$

Then, we have the following final formulation:

$$P\left(\boldsymbol{I} + \frac{1}{2\lambda^*}\nabla^2 C(\boldsymbol{w}^*)\right)^{-1} \tag{38}$$

$$= \frac{1}{\det(M)} \begin{pmatrix} w_z^2(1+B^2 w_y) + Bw_y^2 w_z\tanh(B\Gamma\Delta) & -Bw_y w_z^2\tanh(B\Gamma\Delta) - w_y w_z(1+w_y) \\ -w_y w_z(1+B^2 w_y) - Bw_y^3\tanh(B\Gamma\Delta) & Bw_y^2 w_z\tanh(B\Gamma\Delta) + w_y^2(1+w_y) \end{pmatrix}.$$

**Determining $\boldsymbol{w}^*_{\mathrm{GD}} - \boldsymbol{w}^*$:** We now determine the sign and magnitude of the change in $w_z^*$ when the implicit regularization term induced by Gradient Descent (GD) is applied to the original cost function. By Lemma A.6, we have

$$w^*_{z,\mathrm{GD}} - w_z^* = \frac{-\epsilon}{8\lambda^*}\left[(\boldsymbol{I} - \boldsymbol{w}^*\boldsymbol{w}^{*\top})\left(\boldsymbol{I} + \frac{1}{2\lambda^*}\nabla^2 C(\boldsymbol{w}^*)\right)^{-1}\nabla f(\boldsymbol{w}^*)\right]_z + \mathcal{O}(\epsilon^2)$$

$$= \frac{\epsilon}{8}\frac{4(1+B^2)\bar{\rho}(1-\bar{\rho})e^{-2w_y}}{\frac{e^{-w_y}}{w_y}\sqrt{\bar{\rho}(1-\bar{\rho})}\cosh(B\Gamma\Delta)} w_y \cdot G + \mathcal{O}(\epsilon^2)$$

$$= \frac{\epsilon(1+B^2)\sqrt{\bar{\rho}(1-\bar{\rho})}e^{-w_y}}{2\cosh(B\Gamma\Delta)} w_y^2 \cdot G + \mathcal{O}(\epsilon^2), \tag{39}$$

where $G$ is defined as

$$G \triangleq \frac{G_1 + G_2}{(1+w_y)(1+B^2 w_y) - B^2 w_y^2 \tanh^2(B\Gamma\Delta)},$$

$$G_1 \triangleq -\left(w_z(1+B^2 w_y) + Bw_y^2\tanh(B\Gamma\Delta)\right)\left(\cosh(2B\Gamma\Delta) - \frac{B^2-1}{B^2+1}\right),$$

$$G_2 \triangleq B\left(Bw_y w_z\tanh(B\Gamma\Delta) + w_y(1+w_y)\right)\sinh(2B\Gamma\Delta). \tag{40}$$

Noting $\Gamma \leq 2/3$ due to the assumptions of the theorem, we have $\Delta \ll 1$. Then, applying Taylor expansion (valid for $\Delta \ll 1$) and keeping only first-order terms yields

$$G = -\frac{2w_z}{(B^2+1)(1+w_y)} + \frac{2B^2\Gamma w_y\left(B^2(1+w_y)+1\right)}{(B^2+1)(1+w_y)(1+B^2 w_y)}\Delta. \tag{41}$$

Thus, $G \geq 0$ whenever

$$\Delta \geq \frac{w_z(1+B^2 w_y)}{B^2\Gamma w_y\left(B^2(1+w_y)+1\right)} + \mathcal{O}(B^{-3})$$

$$= \frac{1}{B^2} + \mathcal{O}(B^{-3}). \tag{42}$$

This implies that the reliance on the spurious feature $z$ (measured by $w_{z,\text{GD}}^*$) is strictly larger under GD than under gradient flow. On the other hand, we always have

$$\Delta = \frac{1}{1 + B^2\sqrt{1 - \Gamma^2} + \frac{\Gamma}{1-\Gamma^2}} + \mathcal{O}(B^{-3}) \ \geq \ \frac{1}{B^2} + \mathcal{O}(B^{-3}),$$

for sufficiently large $B$, ensuring that the above condition is satisfied. Finally, note that $G = \mathcal{O}(B^{-2})$ by definition, and

$$\frac{\epsilon(1 + B^2)\sqrt{\bar{\rho}(1 - \bar{\rho})}e^{-w_y}}{2\cosh(B\Gamma\Delta)}\, w_y^2 = \mathcal{O}\!\left(B^2\epsilon\sqrt{\bar{\rho}(1 - \bar{\rho})}\right).$$

Therefore, the unbounded terms with respect to $B$ cancel out, and the proof is complete. $\qquad\square$

### A.3.2 STOCHASTIC GRADIENT DESCENT

This theorem is our main result on gradient descent.

**Theorem A.9.** *[Main Result on Stochastic Gradient Descent for Exponential Loss] Assume the four-point data generation model described in Section A.1.2 with parameters $\rho \in (\frac{1}{100}, \frac{1}{3})$ and $B > 1$. Let $\mathcal{D} = \{(\boldsymbol{X}_i, y_i)\}_{i=1}^n$ be a dataset of $n$ i.i.d. samples drawn from this model. Suppose that*

$$B \ \geq \ \frac{3}{2}\, \log\!\left(\frac{1 - \rho}{\rho}\right), \quad n \ \geq \ \max\left\{ \frac{8b^3 \log(\frac{2}{\zeta})}{(\rho(1 - \rho))^2}, \ \frac{2b \log(\frac{4}{\zeta})}{\rho(1 - \rho)} \right\},$$

*for some $\zeta \in (0, 1)$. Let $w_{z,\text{SGD}}^*$ denote the solution obtained using stochastic gradient descent with a sufficiently small step size $\epsilon > 0$ and minibatch size $b \geq 1$ over a single epoch, and let $w_z^*$ be the solution using gradient flow. Then there exist constants $C_1, C_2 > 0$, depending on $(B, \rho)$ and satisfying $C_1, C_2 = \widetilde{\Theta}(1)$ with respect to both parameters, such that*

$$w_{z,\text{SGD}}^* - w_z^* \ \leq \ C_1\, \epsilon\sqrt{\rho\,(1 - \rho)} - \frac{C_2\, B\epsilon}{b\sqrt{\rho\,(1 - \rho)}} + \mathcal{O}(\epsilon^2 + B^{-1}), \tag{43}$$

*with probability at least $1 - \zeta$ with respect to the randomness of $\mathcal{D}$.*

*Proof of Theorem A.9.* We revisit the derivation for gradient descent, now in the stochastic setting. From Lemma A.4, recall that the surrogate cost for SGD is

$$C_{\text{SGD}}(\boldsymbol{w}) = C(\boldsymbol{w}) + \frac{\epsilon}{4m}\sum_{i=1}^m \left\|\nabla\widehat{C}_i(\boldsymbol{w})\right\|_2^2, \tag{44}$$

where $\widehat{C}_i(\boldsymbol{w})$ is the empirical cost on the $i$th minibatch of size $k = n/m$. If $\rho_i$ denotes the fraction of "bad" samples (those with $z \neq y$) in minibatch $i$, then

$$\widehat{C}_i(\boldsymbol{w}) = \left(\rho_i e^{Bw_z} + (1 - \rho_i)e^{-Bw_z}\right)e^{-w_y}, \tag{45}$$

$$\nabla\widehat{C}_i(\boldsymbol{w}) = \begin{bmatrix} -\widehat{C}_i(\boldsymbol{w}) \\ Be^{-w_y}\left(\rho_i e^{Bw_z} - (1 - \rho_i)e^{-Bw_z}\right) \end{bmatrix}. \tag{46}$$

Consequently,

$$\left\|\nabla\widehat{C}_i(\boldsymbol{w})\right\|_2^2 = \widehat{C}_i^2(\boldsymbol{w}) + B^2 e^{-2w_y}\left(\rho_i^2 e^{2Bw_z} + (1 - \rho_i)^2 e^{-2Bw_z} - 2\rho_i(1 - \rho_i)\right)$$

$$= (1 + B^2)\left(\rho_i^2 e^{2Bw_z} + (1 - \rho_i)^2 e^{-2Bw_z}\right)e^{-2w_y} - 2(B^2 - 1)\rho_i(1 - \rho_i)e^{-2w_y}. \tag{47}$$

Substituting into $C_{\text{SGD}}(\boldsymbol{w})$ yields

$$C_{\text{SGD}}(\boldsymbol{w}) = C(\boldsymbol{w}) + \frac{\epsilon}{4}e^{-2w_y}\left\{(1 + B^2)\left[e^{2Bw_z}\frac{1}{m}\sum_{i=1}^m \rho_i^2 + e^{-2Bw_z}\frac{1}{m}\sum_{i=1}^m (1 - \rho_i)^2\right]\right.$$

$$\left. - 2(B^2 - 1)\frac{1}{m}\sum_{i=1}^m \rho_i(1 - \rho_i)\right\}. \tag{48}$$

Define the minibatch averages

$$\overline{\rho^2} \triangleq \frac{1}{m}\sum_{i=1}^{m}\rho_i^2, \quad \overline{(1-\rho)^2} \triangleq \frac{1}{m}\sum_{i=1}^{m}(1-\rho_i)^2, \quad \overline{\rho(1-\rho)} \triangleq \frac{1}{m}\sum_{i=1}^{m}\rho_i(1-\rho_i).$$

Then $C_{\text{SGD}}$ can be expressed as a perturbation of $C_{\text{GD}}$:

$$C_{\text{SGD}}(\boldsymbol{w}) = C_{\text{GD}}(\boldsymbol{w}) + \frac{\epsilon\,\mathsf{Var}(\rho_{1:m})}{4}e^{-2w_y}\Big((1+B^2)(e^{2Bw_z}+e^{-2Bw_z})+2(B^2-1)\Big), \quad (49)$$

where $\mathsf{Var}(\rho_{1:m}) \triangleq \overline{\rho^2} - \bar{\rho}^2$ is the sample variance across the minibatch proportions.

The remainder of the proof is very similar to that of Theorem A.8. Defining the above residual perturbation term as $\frac{\epsilon}{4}\mathsf{Var}(\rho_{1:m})f(\boldsymbol{w})$, Lemma A.6 applies. Therefore, we need to find how the addition of perturbation term would alter the solution from $\boldsymbol{w}^*\text{GD}$. As in the proof of Theorem A.8, one needs the operator

$$\big(\boldsymbol{I} - \boldsymbol{w}_{\text{GD}}^*\boldsymbol{w}_{\text{GD}}^{*\top}\big)\big(\boldsymbol{I} + \tfrac{1}{2\lambda^*}\nabla^2 C_{\text{GD}}(\boldsymbol{w}_{\text{GD}}^*)\big)^{-1},$$

which suffices at zeroth order in $\epsilon$. The same holds for $\nabla f(\boldsymbol{w}_{\text{GD}}^*)$. Consequently, one can simply consider the $\boldsymbol{w}^*$ and $C(\cdot)$ of the ordinary (non-GD) loss and use the result of Theorem A.8. Carrying out the calculations gives

$$\nabla f(\boldsymbol{w}^*) = -4(1+B^2)e^{-2w_y}\begin{bmatrix}\cosh\big(\log\frac{1-\bar{\rho}}{\bar{\rho}}(1-\Delta)\big)+\frac{B^2-1}{B^2+1}\\-B\sinh\big(\log\frac{1-\bar{\rho}}{\bar{\rho}}(1-\Delta)\big)\end{bmatrix}. \quad (50)$$

After simplification, and noting the fact that due to the assumptions of the theorem we have $w_z^* \le \frac{1}{2B}\log(\frac{1-\bar{\rho}}{rho})$ and $\Delta \ll 1$, this term contributes as

$$\Big[\big(\boldsymbol{I} - \boldsymbol{w}_{\text{GD}}^*\boldsymbol{w}_{\text{GD}}^{*\top}\big)\big(\boldsymbol{I} + \tfrac{1}{2\lambda^*}\nabla^2 C_{\text{GD}}(\boldsymbol{w}_{\text{GD}}^*)\big)^{-1}\nabla f(\boldsymbol{w}^*)\Big]_z = \widetilde{\Theta}\Big(\frac{B}{\bar{\rho}(1-\bar{\rho})}\Big) + \mathcal{O}(\epsilon) + \mathcal{O}(B^{-1}).$$

Applying Lemma A.6, we obtain

$$w_{z,\text{SGD}}^* - w_{z,\text{GD}}^* = -\widetilde{\Theta}\Big(\frac{B\epsilon\,\mathsf{Var}(\rho_{1:m})}{(\bar{\rho}(1-\bar{\rho}))^{3/2}}\Big) + \mathcal{O}(\epsilon^2) + \mathcal{O}(B^{-1}). \quad (51)$$

Finally, Lemma A.7 guarantees that, under the stated conditions on $n$, we have

$$\mathbb{P}\Big(\mathsf{Var}(\rho_{1:m}) \ge \tfrac{\rho(1-\rho)}{2b}\Big) \ge 1 - \zeta,$$

Thus, we have the following with probability at least $1 - \zeta$:

$$w_{z,\text{SGD}}^* - w_z^* \le \widetilde{\Theta}\big(\epsilon\sqrt{\rho(1-\rho)}\big) - \widetilde{\Theta}\Big(\frac{B\epsilon}{b\sqrt{\rho(1-\rho)}}\Big) + \mathcal{O}(\epsilon^2 + B^{-1}), \quad (52)$$

and the proof is complete. □

## A.4 GENERAL RESULTS

We now present a general result that does not rely on the four-point data-generating model of previous results A.1.2 or on any specific choice of loss function or model architecture. Based on the Lemma A.4, we show that in a generic mixture setting with subpopulations overrepresented (majority) and underrepresented (minority), the distributional heterogeneity itself strengthens the implicit regularization induced by SGD. This effect biases the optimization trajectory toward solutions with more uniform performance across subpopulations, thereby reducing the loss gap between Bad solutions and group-robust solutions. In contrast, the implicit regularization induced by GD acts in the opposite direction and amplifies this gap.

Let $\mathcal{D}_{\text{maj}}$ and $\mathcal{D}_{\text{min}}$ denote the majority and minority data distributions, respectively, and let

$$\mathcal{D}_\rho = (1-\rho)\mathcal{D}_{\text{maj}} + \rho\,\mathcal{D}_{\text{min}}, \qquad \rho \in (0,1),$$

be their mixture.

### A.4.1 Stochastic Gradient Descent

In the following theorem, we show that the quantitative analysis carried out for the four-point model under the exponential loss in Section A.3 can be extended—*qualitatively*—to general loss functions (e.g., logistic loss, cross-entropy loss) and general hypothesis classes (e.g., deep neural networks), at least in the asymptotic case where $n, m \to \infty$. Specifically, we consider two solutions, $\boldsymbol{w}_{\text{bad}}$ and $\boldsymbol{w}_{\text{good}}$, where $\boldsymbol{w}_{\text{bad}}$ relies on spurious correlations but attains a *smaller* loss value, while $\boldsymbol{w}_{\text{good}}$ exhibits reduced dependence on spurious features but typically incurs a *larger* loss value (as is common in practice). We prove that the implicit regularization induced by SGD with sufficiently small minibatch size $b$ tends to favor $\boldsymbol{w}_{\text{good}}$ over $\boldsymbol{w}_{\text{bad}}$ by assigning it a smaller regularization value.

The key idea is that hypotheses such as $\boldsymbol{w}_{\text{bad}}$, which rely heavily on spurious correlations, experience significant fluctuations in the number of samples in each minibatch with aligned versus misaligned spurious features. As a result, their minibatch gradients exhibit larger variance—and consequently a larger second moment—which increases the regularization term. In contrast, for the solution $\boldsymbol{w}_{\text{good}}$, these fluctuations are controlled due to its reduced reliance on spurious features and prediction based on core, and therefore leading to a smaller regularization penalty.

**Theorem A.10** (Main Result (Asymptotic) on General Loss Functions and Hypothesis Sets). *Assume any hypothesis set $\mathcal{W}$ (e.g., deep neural networks) and an arbitrary loss function $C(\boldsymbol{w})$ for $\boldsymbol{w} \in \mathcal{W}$. Let $\boldsymbol{w}_{\text{bad}}, \boldsymbol{w}_{\text{good}} \in \mathcal{W}$ be any two hypotheses. In particular, the hypothesis $\boldsymbol{w}_{\text{bad}}$ denotes a minimizer obtained by training on a dataset whose samples exhibit spurious correlations at rate $\rho$, while $\boldsymbol{w}_{\text{good}}$ denotes a group-robust hypothesis whose predictive performance is almost-uniform across both majority and minority subpopulations, yet it is not a minimizer of $C(\boldsymbol{w})$. For some $\varepsilon, \Delta \geq 0$, assume the following conditions hold:*

- $C(\boldsymbol{w}_{\text{bad}}) < C(\boldsymbol{w}_{\text{good}})$,

- $\forall_{i \in m} \big\| \widehat{\boldsymbol{G}}_{\text{maj,i}}^{(\boldsymbol{w}_{\text{good}})} - \widehat{\boldsymbol{G}}_{\text{min,i}}^{(\boldsymbol{w}_{\text{good}})} \big\|_2 \leq \varepsilon$,

- $\forall_{i \in m} \big\| \widehat{\boldsymbol{G}}_{\text{maj,i}}^{(\boldsymbol{w}_{\text{bad}})} - \widehat{\boldsymbol{G}}_{\text{min,i}}^{(\boldsymbol{w}_{\text{bad}})} \big\|_2 \geq \Delta$,

*where $\widehat{\boldsymbol{G}}_{\text{maj},i}^{(\boldsymbol{w})} = \nabla_{\boldsymbol{w}} \widehat{C}_{\text{maj},i}(\boldsymbol{w})$ and $\widehat{\boldsymbol{G}}_{\text{min},i}^{(\boldsymbol{w})} = \nabla_{\boldsymbol{w}} \widehat{C}_{\text{min},i}(\boldsymbol{w})$ denote the average gradients of the majority and minority distributions within the $i - th$ batch, respectively. Define the SGD implicit regularization term (cf. Lemma A.4) as*

$$\mathcal{R}_{\mathcal{D}}^{SGD}(\boldsymbol{w}) \triangleq \frac{\epsilon}{4m} \sum_{k=0}^{m-1} \left\| \nabla \widehat{C}_k(\boldsymbol{w}) \right\|_2^2,$$

*where $m = n/b$ is the number of mini-batches in SGD, and $b$ denotes the batch size. Then, for*

$$b \leq \frac{\rho \, (1 - \rho) \, (\Delta^2 - \varepsilon^2)}{\sup_{\boldsymbol{w}} \|\nabla C(\boldsymbol{w})\|_2^2},$$

*we have*

$$\mathcal{R}_{\mathcal{D}}^{SGD}(\boldsymbol{w}_{\text{bad}}) > \mathcal{R}_{\mathcal{D}}^{SGD}(\boldsymbol{w}_{\text{good}}).$$

*Proof.* We model the random composition of each mini-batch as follows. Let $\alpha_i \sim \text{Binomial}(b, \rho)$ denote the number of minority samples in the $i$-th mini-batch, where $b$ is the batch size and $\rho$ is the minority fraction in the dataset.

The empirical gradient of the $i$-th mini-batch can then be expressed as

$$\boldsymbol{X}_i^{(\boldsymbol{w})} = \alpha_i \widehat{\boldsymbol{G}}_{\text{min,i}}^{(\boldsymbol{w})} + (b - \alpha_i) \widehat{\boldsymbol{G}}_{\text{maj,i}}^{(\boldsymbol{w})} = \alpha_i D^{(\boldsymbol{w})} + b \widehat{\boldsymbol{G}}_{\text{maj,i}}^{(\boldsymbol{w})},$$

where $\widehat{D}_i^{(\boldsymbol{w})} = \widehat{\boldsymbol{G}}_{\text{min,i}}^{(\boldsymbol{w})} - \widehat{\boldsymbol{G}}_{\text{maj,i}}^{(\boldsymbol{w})}$, and $\widehat{\boldsymbol{G}}_{\text{min,i}}, \widehat{\boldsymbol{G}}_{\text{maj,i}}$ are the average of gradients of minority and majority samples in the $i - th$ batch, respectively. Using Lemma A.4 and taking the large-sample limit $n, m \to \infty$, the expected implicit regularization term becomes

$$\mathcal{R}_{\mathcal{D}}^{SGD}(\boldsymbol{w}) = \frac{\epsilon}{4} \, \mathbb{E} \big\| \boldsymbol{X}_i^{(\boldsymbol{w})} \big\|_2^2.$$

Let us define

$$\boldsymbol{\xi}_i^{(\boldsymbol{w})} \triangleq (\alpha_i - \rho b)\widehat{D}_i^{(\boldsymbol{w})}, \qquad \boldsymbol{\mu}^{(\boldsymbol{w})} \triangleq b\big(\rho\widehat{\boldsymbol{G}}_{\mathrm{min,i}}^{(\boldsymbol{w})} + (1 - \rho)\widehat{\boldsymbol{G}}_{\mathrm{maj,i}}^{(\boldsymbol{w})}\big).$$

Then we have the decomposition

$$\boldsymbol{X}_i^{(\boldsymbol{w})} = \boldsymbol{\mu}^{(\boldsymbol{w})} + \boldsymbol{\xi}_i^{(\boldsymbol{w})}.$$

By construction,

$$\mathbb{E}[\boldsymbol{\xi}_i^{(\boldsymbol{w})}] = 0, \qquad \mathrm{Var}(\alpha_i) = b\rho(1 - \rho), \qquad \mathbb{E}[\|\boldsymbol{\xi}_i\|_2^2] = b\rho(1 - \rho)\|\widehat{D}_i^{(\boldsymbol{w})}\|_2^2.$$

Using the zero–mean property of $\boldsymbol{\xi}_i$,

$$\mathbb{E}\|\boldsymbol{X}_i\|_2^2 = \|\boldsymbol{\mu}\|_2^2 + \mathbb{E}\|\boldsymbol{\xi}_i\|_2^2.$$

Substituting the definitions yields

$$\mathbb{E}\|\boldsymbol{X}_i^{(\boldsymbol{w})}\|_2^2 = \left\|b\big(\rho\boldsymbol{G}_{\mathrm{min}}^{(\boldsymbol{w})} + (1 - \rho)\boldsymbol{G}_{\mathrm{maj}}^{(\boldsymbol{w})}\big)\right\|_2^2 + b\rho(1 - \rho)\|\widehat{D}_i^{(\boldsymbol{w})}\|_2^2 \tag{53}$$

$$= b^2\left\|\nabla C(\boldsymbol{w})\right\|_2^2 + b\rho(1 - \rho)\|\widehat{D}_i^{(\boldsymbol{w})}\|_2^2 \tag{54}$$

Substituting into $\mathcal{R}_{\mathcal{D}}^{SGD}(\boldsymbol{w}_{bad}) - \mathcal{R}_{\mathcal{D}}^{SGD}(\boldsymbol{w}_{good})$ we have

$$\mathcal{R}_{\mathcal{D}}^{SGD}(\boldsymbol{w}_{bad}) - \mathcal{R}_{\mathcal{D}}^{SGD}(\boldsymbol{w}_{good}) = \frac{\epsilon}{4}\left[\|\boldsymbol{X}_i^{(\boldsymbol{w}_{bad})}\|_2^2 - |\boldsymbol{X}_i^{(\boldsymbol{w}_{good})}\|_2^2\right] \tag{55}$$

$$= \frac{\epsilon\, b^2}{4}\left[\left\|\nabla C(\boldsymbol{w}_{bad})\right\|_2^2 - \left\|\nabla C(\boldsymbol{w}_{good})\right\|_2^2 + \right. \tag{56}$$

$$\left. \frac{\rho(1 - \rho)}{b}\big(\|\widehat{D}_i^{(\boldsymbol{w}_{bad})}\|_2^2 - \|\widehat{D}_i^{(\boldsymbol{w}_{good})}\|_2^2\big)\right] \tag{57}$$

$$\geq \frac{\epsilon\, b^2}{4}\left[\left\|\nabla C(\boldsymbol{w}_{bad})\right\|_2^2 - \left\|\nabla C(\boldsymbol{w}_{good})\right\|_2^2 + \right. \tag{58}$$

$$\left. \frac{\rho(1 - \rho)}{b}\big(\Delta^2 - \varepsilon^2\big)\right] \tag{59}$$

Since $\boldsymbol{w}_{\mathrm{bad}}$ is a stationary point of $C$, we have $\|\nabla C(\boldsymbol{w}_{bad})\| = 0$. Using condition on $b$ we have:

$$\mathcal{R}_{\mathcal{D}}^{SGD}(\boldsymbol{w}_{bad}) - \mathcal{R}_{\mathcal{D}}^{SGD}(\boldsymbol{w}_{good}) \geq \frac{\epsilon\, b^2}{4}\left[-\left\|\nabla C(\boldsymbol{w}_{good})\right\|_2^2 + \sup_{\boldsymbol{w}}\left\|\nabla C(\boldsymbol{w})\right\|_2^2\right] \geq 0 \tag{60}$$

This establishes that

$$\mathcal{R}_{\mathcal{D}}^{SGD}(\boldsymbol{w}_{\mathrm{bad}}) \geq \mathcal{R}_{\mathcal{D}}^{SGD}(\boldsymbol{w}_{\mathrm{good}}),$$

and therefore completes the proof.

$$\square$$

### A.4.2 GRADIENT DESCENT

**Theorem A.11** (General Result on the Implicit Regularization of GD). *Consider two parameter vectors $\boldsymbol{w}_{\mathrm{bad}}, \boldsymbol{w}_{\mathrm{good}} \in \mathbb{R}^d$. The solution $\boldsymbol{w}_{\mathrm{bad}}$ denotes a minimizer obtained by training on a dataset whose samples exhibit spurious correlations at rate $\rho$, while $\boldsymbol{w}_{\mathrm{good}}$ denotes a group-robust classifier whose predictive performance is uniform across both majority and minority subpopulations.*

*Assume that the corresponding population losses satisfy*

$$C(\boldsymbol{w}_{\mathrm{bad}}) < C(\boldsymbol{w}_{\mathrm{good}}).$$

*Define the GD implicit regularization term (cf. Lemma A.3) as*

$$\mathcal{R}_{\mathcal{D}}^{GD}(\boldsymbol{w}) \triangleq \frac{\epsilon}{4}\left\|\nabla C(\boldsymbol{w})\right\|_2^2,$$

*Then, we have*

$$\mathcal{R}_{\mathcal{D}}^{GD}(\boldsymbol{w}_{\mathrm{bad}}) < \mathcal{R}_{\mathcal{D}}^{GD}(\boldsymbol{w}_{\mathrm{good}}).$$

*Proof.* Since $\boldsymbol{w}_{\text{bad}}$ is a minimizer of $C$, it is a stationary point and therefore satisfies $\nabla C(\boldsymbol{w}_{\text{bad}}) = 0$. Consequently,

$$\mathcal{R}_{\mathcal{D}}^{GD}(\boldsymbol{w}_{bad}) - \mathcal{R}_{\mathcal{D}}^{GD}(\boldsymbol{w}_{good}) = \frac{\epsilon}{4}\left[\left\|\nabla C(\boldsymbol{w}_{bad})\right\|_2^2 - \left\|\nabla C(\boldsymbol{w}_{good})\right\|_2^2\right] \tag{61}$$

$$= -\frac{\epsilon}{4}\left\|\nabla C(\boldsymbol{w}_{\text{good}})\right\|_2^2 < 0, \tag{62}$$

which establishes the claim. $\qquad\square$

Theorems A.10 and A.11 show that, for any cost function $C$ and any model architecture, the mere presence of gradient discrepancies between samples in the training data is sufficient to increase the implicit regularization effect of SGD. This effect acts to reduce the loss gap between shortcut and group-robust solutions. Our findings align with the general understanding of implicit regularization of SGD that smooths the loss landscape by penalizing gradient variance across samples, thereby favoring solutions with more uniform loss across samples. In contrast, full-batch GD lacks this mechanism and, in our setting, exhibits the opposite behavior, amplifying rather than reducing the gap between bad and good solutions.

# B EXPERIMENTS

## B.1 DATASETS

In our experiments, we evaluate models on a diverse set of datasets that are specifically designed to test robustness against spurious correlations. Each dataset introduces a known, controllable spurious feature that can confound standard training methods. Below, we briefly describe the datasets used in our experiments. Some example images from each dataset, including both majority and minority samples, are shown in Table 4.

- **Waterbirds** (Sagawa et al., 2020a) is a synthetic dataset generated by placing bird images from the CUB dataset onto backgrounds from the Places dataset. The task is binary classification: waterbird versus landbird. In this dataset, the background type (water or land) is strongly correlated with the bird type, creating a pronounced bias—most waterbirds appear on water backgrounds, and most landbirds on land. Consequently, models trained normally often rely on the background instead of bird-specific features, which reduces generalization performance for minority groups where this correlation is reversed.

- **CelebA** (Liu et al., 2015) contains over 200,000 celebrity face images annotated with 40 binary attributes (e.g., smiling, wearing glasses, hair color) along with identity labels. It presents a multi-label classification challenge, as each image can have multiple attributes simultaneously. The dataset exhibits natural biases in attribute co-occurrence and demographic distributions—for example, blonde hair is far more common among women than men. As a result, standard models may rely on hair color as a shortcut for predicting gender, failing to generalize to minority groups where this correlation does not hold. CelebA thus serves as a valuable benchmark for testing methods that aim for robust and fair facial attribute prediction.

- **CIFAR-10 (Car vs. Truck)** Lubana et al. (2023) is a subset of the CIFAR-10 dataset (Krizhevsky et al., 2009) limited to two classes: car and truck. To create a spurious correlation, a small colored square is added to the top-left corner of each image. The square's color is strongly associated with the label (e.g., green for cars, pink for trucks). A small portion of samples breaks this correlation by having an opposite or random color. This setup allows evaluation of whether models rely on the spurious cue or on the object's true shape for classification.

- **Cmnist** Arjovsky et al. (2019) is a variant of the MNIST dataset with 10 classes corresponding to the digits 0 through 9. Each grayscale digit is assigned a color determined primarily by its label (e.g., $0 \rightarrow$ red, $1 \rightarrow$ green), while a small proportion is colored randomly to introduce variation. This setup creates a strong but spurious correlation between digit identity and color, even though the true predictive signal is the digit shape. As a result,

standard models often over-rely on color, leading to reduced performance when the correlation changes at test time. Cmnist thus serves as a simple and interpretable benchmark for evaluating robustness to spurious features.

- **Cmnist2** is a binary-class variant of Cmnist constructed using only the digits 0 and 1. It follows the same coloring scheme as Cmnist, but restricts the task to distinguishing between two classes, providing a simpler setting for studying robustness methods.

- **Dominoes** Murali et al. (2023) is a synthetic dataset designed to examine model behavior under multiple potential spurious features. It pairs CIFAR-10 images with Fashion-MNIST images of the same class (e.g., a "cat" image with a "pullover"), forming composite images. The CIFAR-10 segment serves as the primary cue, while the Fashion-MNIST segment introduces a structured but potentially spurious feature. Models may preferentially use the easier-to-learn Fashion-MNIST portion during training. Domino supports controlled interventions like removing or randomizing the Fashion-MNIST side, making it ideal for studying spurious feature learning and robustness.

- **MultiNLI** Williams et al. (2018) is a dataset where each sentence pair is labeled as entailment, neutral, or contradiction. The dataset contains many examples of these three types of relationships. We use a spurious feature from Sagawa et al. (2020a), which is the presence of negation words in the second sentence. Because of the way the data was collected, sentences labeled as contradictions often contain negation words.

- **CivilComments-WILDS** (Borkan et al., 2019; Koh et al., 2021a) is a dataset designed for the task of classifying online comments as toxic or non-toxic. In this dataset, the labels are spuriously correlated with mentions of specific demographic identities. Following the evaluation protocol of (Koh et al., 2021a), we consider 16 overlapping groups—each demographic identity paired with toxic or non-toxic labels.

## B.2 EXPERIMENTAL SETUP

For architectures, we used the PyTorch implementations of ResNet-50 (He et al., 2016) for Waterbirds, ResNet-18 for CIFAR-10 Domino, and CelebA and a three-layer MLP with two hidden layers of 128 units for Colored MNIST. ResNet backbones were initialized with ImageNet-pretrained weights; inputs were normalized using the ImageNet mean and standard deviation. Colored MNIST images were normalized with a mean and standard deviation of 0.5 across all channels. For fair comparison of performance under small and large batch sizes, BatchNorm layers in ResNets were replaced with GroupNorm (Wu & He, 2018) using 32 groups as suggested in Smith et al. (2021). For the MultiNLI and CivilComments datasets, we used BERT (Devlin et al., 2019a). We applied the HuggingFace implementation of BERT (Wolf et al., 2020) and started from the pretrained model weights.

All models were optimized with SGD with momentum $0.9$ and weight decay $10^{-5}$ and learning rates $\{10^{-6}, 10^{-5}, \ldots, 10^{-1}\}$. All ResNet models were trained for 300 epochs; the Colored MNIST MLP was trained for 150 epochs.

All reported results for effect of learning rate and batch size on WGA and ACC are reported by final model (without model selection based on validation set). In contrast, when analyzing the correlation between the normalized SGD implicit regularization term and WGA, we perform model selection using validation-set ACC. This approach allows us to approximate the choice of parameters corresponding to local minima.

We used an NVIDIA GeForce RTX 4090 for all runs, except Waterbirds runs with batch sizes of 128 and 256, which were executed on an NVIDIA A100 (80 GB) to accommodate the memory required to compute the implicit regularization term at the end of training across all splits.

Results are reported as mean $\pm$ standard deviation over three independent random seeds; seeds affect dataset generation and model initialization.

## B.3 EXPLICIT DEBIASING METHODS

**AFR** (Qiu et al., 2023) first trains a model using standard ERM, and then retrains the classifier on a weighted held-out dataset. The weights for each sample are based on the probability that the ERM-

pretrained model assigns to the correct label, effectively giving more importance to samples from minority groups. This approach aims to reduce bias by emphasizing underrepresented groups during the retraining phase.

**DFR** (Kirichenko et al., 2023) assumes that ERM-trained models are capable of capturing the core, invariant features of the data. It first trains the full model with ERM, and then retrains only the last linear classifier layer using a group-balanced subset of the validation set or held-out training data. While DFR minimizes the need for extensive group annotations, it still requires group labels for the retraining step.

**EVaLS** (Ghaznavi et al., 2025) (Environment-based Validation and Loss-based Sampling) removes the need for group annotations entirely. It leverages the loss values of an ERM-trained model to identify hard or misclassified samples and constructs a balanced held-out dataset for last-layer retraining. This approach improves robustness to spurious correlations by using high-loss samples as proxies for underrepresented groups, effectively achieving group robustness without explicit group labels.

### B.4 Optimal WGA and ACC per Batch Size

Across all datasets, worst-group accuracy (WGA) exhibits a substantially larger drop with increasing batch size compared to overall accuracy (ACC). This indicates that once in-distribution generalization is saturated, implicit regularization continues to drive out-of-distribution gains by enhancing group robustness. In particular, smaller batch sizes tend to achieve the highest WGA, whereas larger batches often degrade it, even when ACC remains nearly unchanged. The consistent gap between $\Delta$WGA and $\Delta$ACC confirms that improvements in robustness cannot be solely attributed to better average accuracy, but rather to the differential inductive biases induced by the optimization dynamics (Table 5).

### B.5 Effect of Batch Size and Learning Rate on WGA and ACC

Across vision datasets, overall accuracy (ACC) quickly saturates as the learning rate increases, while worst-group accuracy (WGA) continues to benefit from stronger implicit regularization (Table 6). This effect is particularly pronounced in biased datasets, where higher learning rates substantially improve WGA even after ACC has plateaued. The contrast between biased and balanced settings highlights that optimization dynamics influence robustness more strongly than average accuracy. A similar trend is observed in the text domain on the MultiNLI dataset (Table 6), indicating that the relationship between learning rate, batch size, and worst-group robustness generalizes beyond vision tasks.

## C Large Language Model (LLM) Usage Disclosure

We used large language models (LLMs) only for editing support (clarity/grammar) and for minor non-core code cleanup (formatting, docstrings, renaming, etc.). All LLM-assisted edits were reviewed by the authors, who remain fully responsible under the ICLR 2026 policy.

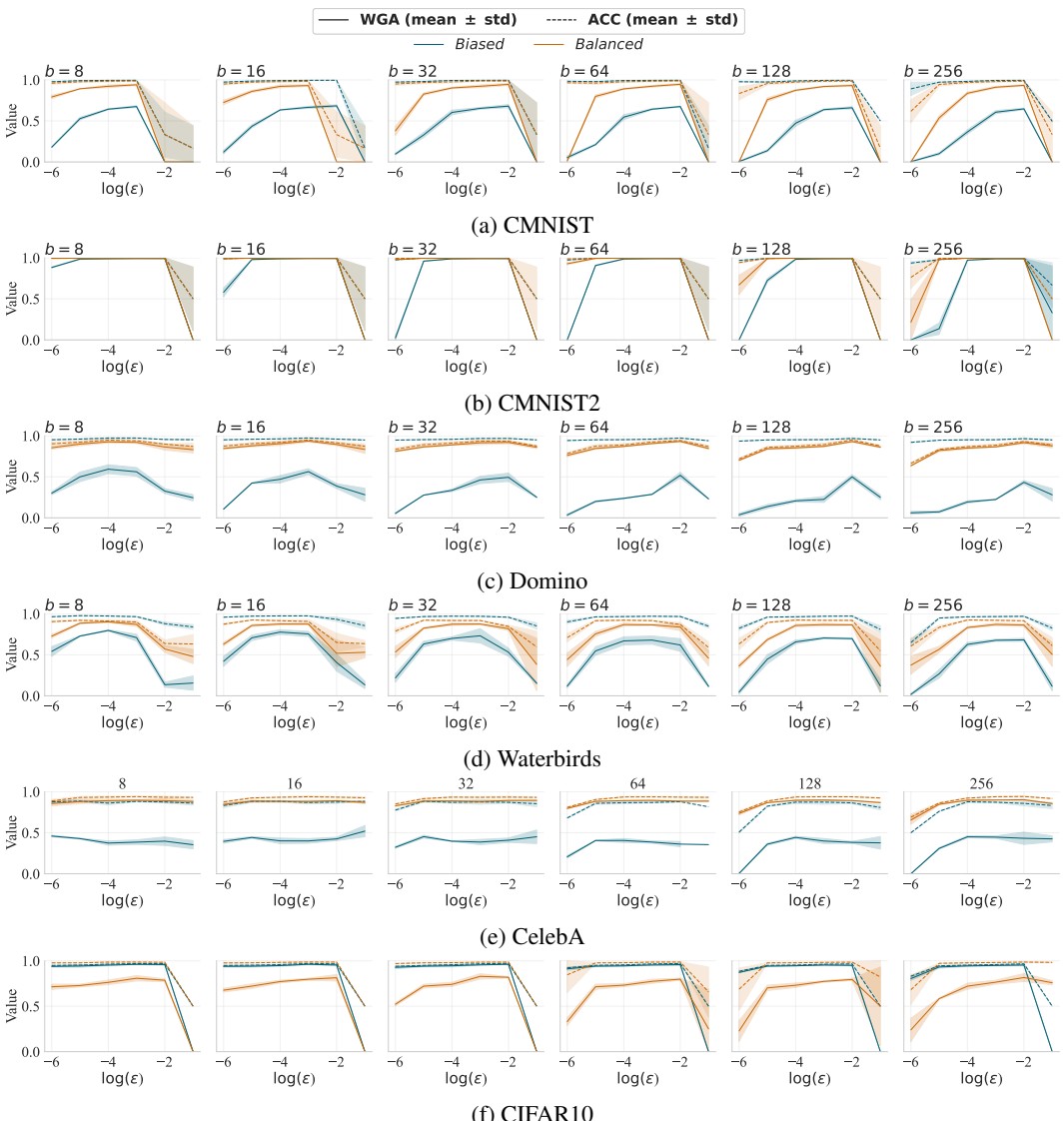

Figure 6: **WGA and ACC across batch sizes** ($b$) **and learning rates** ($\epsilon$) **for biased** ($\rho = 5\%$) **and balanced datasets.** After ACC saturates (dashed lines), increasing the learning rate substantially boosts WGA (solid lines). The effect is stronger in biased datasets (orange) than in balanced ones (blue).

Table 4: Representative examples of majority and minority groups from the benchmark datasets (Waterbirds, CelebA, C-MNIST, CIFAR-10, and Domino). For each dataset, majority and minority samples are shown side by side to illustrate the spurious feature: background in Waterbirds, hair color in CelebA, digit color in C-MNIST, small square in the left corner in CIFAR-10, and number position in the lower part of each image in the Domino dataset.

| Dataset | Group | Examples |
|---------|-------|----------|
| Waterbirds | *Majority* |  |
| | *Minority* |  |
| CelebA | *Majority* |  |
| | *Minority* |  |
| CIFAR10 | *Majority* |  |
| | *Minority* |  |
| Cmnist | *Majority* |  |
| | *Minority* |  |
| Domino | *Majority* |  |
| | *Minority* |  |

Table 5: Optimal learning rates and their corresponding WGA and ACC on the test set, reported as mean$_{\pm\text{std}}$ (%) for each batch size across biased datasets with spurious correlation $\rho = 5\%$. The highest and lowest values for each dataset are highlighted in blue and yellow, respectively. At the end of each dataset block, $\Delta$ is defined as the difference between the maximum and minimum values. The magnitude of changes in WGA exceeds that of ACC across all datasets, suggesting that improvements in group robustness are not merely a byproduct of in-distribution generalization enhancement achieved through strong implicit regularization.

**CMNIST**

| $b$ | $\epsilon^\star$ | $WGA^\star$ | $ACC$ |
|---|---|---|---|
| 8 | $10^{-3}$ | $67.5_{\pm1.1}$ | $99.5_{\pm0.0}$ |
| 16 | $10^{-2}$ | $68.4_{\pm1.2}$ | $99.5_{\pm0.0}$ |
| 32 | $10^{-2}$ | $68.0_{\pm1.9}$ | $99.6_{\pm0.0}$ |
| 64 | $10^{-2}$ | $67.5_{\pm0.6}$ | $99.5_{\pm0.0}$ |
| 128 | $10^{-2}$ | $66.0_{\pm1.3}$ | $99.5_{\pm0.0}$ |
| 256 | $10^{-2}$ | $64.7_{\pm0.9}$ | $99.5_{\pm0.0}$ |
| $\Delta$ | | **+3.7** | **+0.1** |

**CIFAR10**

| $b$ | $\epsilon^\star$ | $WGA^\star$ | $ACC$ |
|---|---|---|---|
| 8 | $10^{-3}$ | $80.1_{\pm2.2}$ | $98.7_{\pm0.1}$ |
| 16 | $10^{-3}$ | $78.9_{\pm1.7}$ | $98.8_{\pm0.1}$ |
| 32 | $10^{-3}$ | $79.6_{\pm1.7}$ | $98.9_{\pm0.2}$ |
| 64 | $10^{-2}$ | $78.8_{\pm2.8}$ | $98.8_{\pm0.1}$ |
| 128 | $10^{-2}$ | $76.3_{\pm2.8}$ | $98.6_{\pm0.2}$ |
| 256 | $10^{-2}$ | $77.9_{\pm2.5}$ | $98.7_{\pm0.2}$ |
| $\Delta$ | | **+3.8** | **+0.3** |

**Waterbirds**

| $b$ | $\epsilon^\star$ | $WGA^\star$ | $ACC$ |
|---|---|---|---|
| 8 | $10^{-4}$ | $79.7_{\pm0.8}$ | $97.3_{\pm0.5}$ |
| 16 | $10^{-4}$ | $77.7_{\pm2.8}$ | $97.4_{\pm0.3}$ |
| 32 | $10^{-3}$ | $73.2_{\pm8.7}$ | $96.9_{\pm0.6}$ |
| 64 | $10^{-3}$ | $67.9_{\pm5.0}$ | $97.0_{\pm0.3}$ |
| 128 | $10^{-3}$ | $70.4_{\pm0.9}$ | $97.0_{\pm0.2}$ |
| 256 | $10^{-2}$ | $68.2_{\pm1.9}$ | $96.9_{\pm0.4}$ |
| $\Delta$ | | **+11.5** | **+0.5** |

**Domino**

| $b$ | $\epsilon^\star$ | $WGA^\star$ | $ACC$ |
|---|---|---|---|
| 8 | $10^{-4}$ | $59.3_{\pm5.3}$ | $97.2_{\pm0.5}$ |
| 16 | $10^{-3}$ | $56.3_{\pm3.6}$ | $97.4_{\pm0.3}$ |
| 32 | $10^{-2}$ | $49.4_{\pm5.1}$ | $96.8_{\pm0.3}$ |
| 64 | $10^{-2}$ | $51.9_{\pm3.6}$ | $97.2_{\pm0.3}$ |
| 128 | $10^{-2}$ | $50.0_{\pm2.6}$ | $96.8_{\pm0.4}$ |
| 256 | $10^{-2}$ | $43.4_{\pm2.1}$ | $96.5_{\pm0.4}$ |
| $\Delta$ | | **+15.9** | **+0.9** |

**CMNIST2**

| $b$ | $\epsilon^\star$ | $WGA^\star$ | $ACC$ |
|---|---|---|---|
| 8 | $10^{-3}$ | $99.1_{\pm0.2}$ | $100.0_{\pm0.0}$ |
| 16 | $10^{-3}$ | $99.2_{\pm0.2}$ | $100.0_{\pm0.0}$ |
| 32 | $10^{-3}$ | $99.1_{\pm0.2}$ | $100.0_{\pm0.0}$ |
| 64 | $10^{-3}$ | $99.0_{\pm0.3}$ | $100.0_{\pm0.0}$ |
| 128 | $10^{-2}$ | $99.0_{\pm0.2}$ | $100.0_{\pm0.0}$ |
| 256 | $10^{-3}$ | $98.9_{\pm0.1}$ | $100.0_{\pm0.0}$ |
| $\Delta$ | | **+0.3** | **+0.0** |

**CelebA**

| $b$ | $\epsilon^\star$ | $WGA^\star$ | $ACC$ |
|---|---|---|---|
| 8 | $10^{-6}$ | $46.0_{\pm1.0}$ | $87.6_{\pm1.2}$ |
| 16 | $10^{-1}$ | $51.9_{\pm6.8}$ | $88.2_{\pm0.6}$ |
| 32 | $10^{-5}$ | $45.3_{\pm1.8}$ | $88.4_{\pm1.2}$ |
| 64 | $10^{-4}$ | $40.5_{\pm2.4}$ | $86.6_{\pm1.2}$ |
| 128 | $10^{-4}$ | $44.3_{\pm1.3}$ | $87.2_{\pm1.6}$ |
| 256 | $10^{-4}$ | $45.0_{\pm1.3}$ | $87.7_{\pm1.3}$ |
| $\Delta$ | | **+11.4** | **+1.8** |

Table 6: Effect of batch sizes and learning rates on WGA and ACC for the **Multi-NLI** dataset.

| $b$ | $\varepsilon = 10^{-3}$ | | $\varepsilon = 10^{-4}$ | | $\varepsilon = 10^{-5}$ | |
|---|---|---|---|---|---|---|
| | WGA | ACC | WGA | ACC | WGA | ACC |
| 8 | $78.24_{\pm0.56}$ | $81.25_{\pm0.22}$ | $76.75_{\pm0.48}$ | $82.18_{\pm0.04}$ | $76.88_{\pm1.14}$ | $81.72_{\pm0.09}$ |
| 16 | $77.64_{\pm0.89}$ | $81.29_{\pm0.03}$ | $76.58_{\pm2.04}$ | $82.40_{\pm0.07}$ | $76.31_{\pm0.83}$ | $81.31_{\pm0.63}$ |
| 32 | $77.45_{\pm1.21}$ | $82.12_{\pm0.61}$ | $76.50_{\pm0.57}$ | $81.74_{\pm0.20}$ | $76.73_{\pm0.88}$ | $81.59_{\pm0.13}$ |
| 64 | $77.28_{\pm1.95}$ | $81.81_{\pm0.21}$ | $75.80_{\pm1.85}$ | $81.93_{\pm0.13}$ | $74.57_{\pm0.70}$ | $79.46_{\pm0.09}$ |
| 128 | $77.73_{\pm0.86}$ | $81.86_{\pm0.18}$ | $75.78_{\pm0.68}$ | $80.96_{\pm0.05}$ | $73.92_{\pm0.47}$ | $78.33_{\pm0.20}$ |
| 256 | $76.74_{\pm0.48}$ | $81.67_{\pm0.27}$ | $75.17_{\pm0.63}$ | $79.34_{\pm0.02}$ | $71.37_{\pm0.24}$ | $77.93_{\pm0.19}$ |

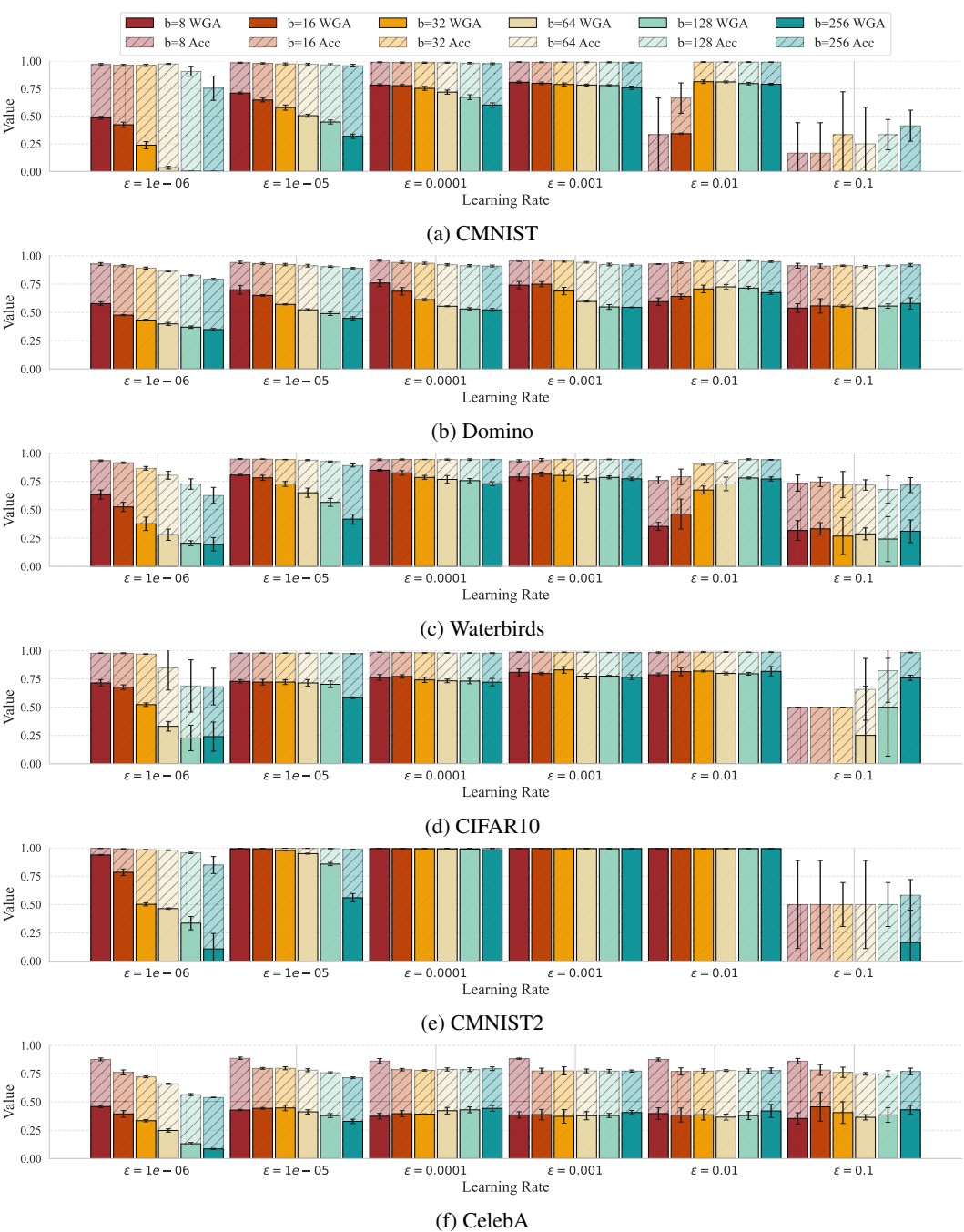

Figure 7: **Joint Effect of Learning Rate and Batch Size on WGA and ACC.** Once the learning rate is sufficiently large to guarantee ACC, smaller batch sizes consistently yield higher WGA across all datasets, indicating improved robustness. It is important to note, however, that for very high or very low learning rates, some datasets fail to achieve in-distribution generalization (e.g., $\epsilon = 0.1$ across all datasets or $\epsilon = 10^{-6}$ for Waterbirds). In such cases, reliable conclusions regarding robustness cannot be drawn.

