# OpenReview forum: "Implicit Regularization of SGD Reduces Shortcut Learning"
_ICLR.cc/2026/Conference — ICLR 2026 Poster_

### Official Review · Reviewer_AkY5 · 2025-10-29

**Soundness:** 3
**Presentation:** 3
**Contribution:** 3
**Rating:** 6
**Confidence:** 3

**Summary:**

The paper studies the effect of batch size on the reliance of stochastic gradient descent on spurious features. The paper considers a simple 4 point model where there are two binary features, one exactly equal to the label and one spurious feature with high correlation with the label  and the spurious one is also larger in magnitude. The paper studies the solution for this problem returned by gradient descent and stochastic gradient descent with an exponential loss function. It shows that due to implicit bias, gradient descent increases the coefficient of the spurious feature whereas stochastic gradient descent with small batch sizes reduces the coefficient of the spurious feature.

**Strengths:**

The paper demonstrates an interesting phenomenon relating the implicit biases of different algorithms to their robustness to spurious features. I am not very familiar with the literature on implicit biases but I have not seen this relation being explored before.

The experiments show that the conclusions also hold to some extent on realistic cases with standard datasets. For a fixed batch size, increasing step size appears to improve the worst group error. For a fixed step size, reducing batch size also appears to improve the worst group error (only if the average error is maintained).

**Weaknesses:**

The previous works by Puli et al. and Sagawa et al. cited by the paper use the logistic loss function whereas this paper uses an exponential loss function. It would be helpful to explain this change and whether the conclusion continues to hold in the same setting as previous works.

The conclusions for GD and SGD also need not hold for more sophisticated methods for debiasing, as described in section 4.4.

The amount of robustness provided by SGD with small batch size seems very small compared with more sophisticated methods in the experiments. If one's goal is to achieve robustness, other methods might be the major factors and the batch size a minor one.

**Questions:**

Could you please explain the reason for changing the loss function.

---

> ### Author Response · Authors · 2025-11-21
> **Authors’ Response to Reviewer AkY5**
>
> Thank you for your constructive feedback, and positive recommendation to accept our work. We address each comments below, and we would be happy to provide further clarification on any remaining concerns.
>
> ---
> ### ❖ Reviewer Comment 1
> We thank you for raising this point, as it highlights an important conceptual distinction between our work and prior analyses.
>
> A large body of prior work shows that gradient-based optimizers trained on linearly separable data with a broad class of monotonically decreasing, smooth margin-based loss functions (including logistic, exponential, and probit) converge to the $\ell_2$​ max-margin classifier in the limit of infinite iterations [1,2,3,4]. Formally
> $$
> 	\lim_{t \to \infty} \frac{w(t)}{\|w(t)\|} = \frac{\hat{w}}{\|\hat{w}\|}
> \tag{1}
> $$
> where $\hat{w}$ is the hard-margin SVM solution. Because this result is loss-agnostic, prior works such as [5, 6, 7] rely on (1) regardless of their choice of loss function.
>
> However, our goal is different: we study the optimization trajectory, and specifically how batch size and learning rate modulate the reliance of spurious vs. invariant weights. These dynamics cannot be captured by the formula (1) and require analyzing the gradients and curvature of the loss.
>
> We chose the exponential loss for its mathematical tractability, which enables closed-form derivations that are not equivalently available under the logistic loss. This yields clearer derivations and avoids obscuring the main conceptual message. For the minority group, when B is large, the exponential and logistic losses behave similarly. For the majority group, the logistic loss is strictly smaller than the exponential loss in the large B regime, which would only reduce the optimal reliance on the spurious component. However, the effect of implicit regularization on shortcut reliance, **does not rely on a property specific to the exponential function**, hence, it is natural to expect the results to extend to the logistic loss. Using the exponential loss simply yields clearer derivations and avoids obscuring the main conceptual message.
>
> Consistent with this, we want to emphasise that
> 1) **Our formal results rely on the difference between the implicit regularization terms of GD and SGD, not on any special property of the exponential loss.** SGD penalizes mini-batch gradient variance, whereas GD does not. Any loss function that induces a max-margin-like solution for separable data can still encourage reliance on the feature with the larger margin (the spurious feature), leading to higher gradient variance across mini-batches. This mechanism does not depend on the exponential loss.
> 2) **All of our deep-network experiments use cross-entropy loss**, yet exhibit behavior consistent with our theoretical predictions. This provides strong empirical evidence that the mechanism we study—implicit regularization and its effect on shortcut reliance—is not restricted to the exponential loss.
>
> We agree that extending our analysis to other losses like logistic loss, would be an interesting direction for this work.
>
> ---
> ### ❖ Reviewer Comment 2
> We fully agree that in **single-level spurious datasets such as CelebA and Waterbirds, training debiasing methods with small batch sizes is not necessarily beneficial**, and in some cases, may even degrade performance.
>
> Small-batch ERM tends to encourage the model to rely more on core features, thereby producing representations that are inherently more robust. However, two-stage methods such as DFR perform last-layer retraining on balanced, group-annotated data, which can effectively eliminate reliance on spurious features. Therefore, reducing the batch size during ERM, which aims to promote robustness and reduce dependence on spurious features, does not necessarily benefit these methods, as they already address this issue in the second stage via last-layer retraining.
>
> Conversely, methods like AFR and EVaLS rely on group discovery, **which depends on a classifier trained on representations that encode spurious correlations.** As a result, when the base model relies more heavily on core features, as occurs with small batch sizes, the quality of their group discovery deteriorates. This explains why AFR and EVaLS exhibit performance degradation under small batch sizes, as shown in Table 2 in  Waterbirds and CelebA.
>
> Importantly, **the situation changes in multi-level spurious datasets such as Domino-CMF**. In this more realistic and challenging setting, even debiasing methods benefit substantially from small-batch training yielding 25–37% improvements in WGA. When spurious attributes are unknown or complex, small-batch ERM provides implicit robustness without requiring any group labels or prior knowledge of the spurious structure, making it a practical trick for scenarios where debiasing methods struggle.
>
> ---
> ### ❖ Reviewer Comment 3
> About practical usefulness, please refer to our common response.

---

> > ### Author Response · Authors · 2025-11-21
> > **References**
> >
> > ### References
> >
> > [1] Soudry et al. The implicit bias of gradient descent on separable data. JMLR 2018
> >
> > [2] Nacson et al. Stochastic gradient descent on separable data: Exact convergence with a fixed learning rate. AISTATS 2019
> >
> > [3] Lyu & Li. Gradient descent maximizes the margin of homogeneous neural networks. ICLR 2020
> >
> > [4] Chizat & Bach. Implicit bias of gradient descent for wide two-layer neural networks. arXiv 2020
> >
> > [5] Sagawa et al. An investigation of why overparameterization exacerbates spurious correlations. ICML 2020
> >
> > [6] Puli et al. Don’t blame dataset shift! Shortcut learning due to gradients and cross entropy. NeurIPS 2023
> >
> > [7] Nagarajan et al. Understanding the failure modes of out-of-distribution generalization. ICLR 2021

---

> > ### Comment · Reviewer_AkY5 · 2025-11-25
> > **Acknowledgment**
> >
> > I thank the authors for the responses. I continue to think that the provided theory on a simple model gives an explanation for the relationship between the batch size and the robustness, which is an interesting phenomenon. However, even if one is fine with the very simplistic theory model, previous works hold for all reasonable loss functions including the one used in the experiments, the new theory narrowly holds for a different loss function from the experiments. I also continue to think that the improvement from varying the batch size is relatively small compared with the robustness requirement or what can be obtained by other methods. Thus, I am still slightly positive about the paper but cannot fully endorse it.

---

> ### Author Response · Authors · 2025-11-26
>
> We thank the reviewer for raising the concern that "previous works hold for all reasonable loss functions, including the one used in the experiments, the new theory narrowly holds for a different loss function from the experiments." To address this, we provide two pieces of evidence.
>
> 1) **Empirical validation with exponential loss.** We reproduce our empirical results with the exponential loss CMNIST dataset. For this experiment we use a one-vs-all exponential loss $L(x,y;f)=\sum_{k=1}^K \exp(-y_k f_k(x))$, where $y_k\in{-1,+1}$ with $+1$ for the ground-truth class and $-1$ otherwise [1]. As expected, within the regime where in-distribution generalization (measured by ACC) remains near optimal, increasing the learning rate and reducing the batch size again lead to improved worst-group accuracy (WGA), with the same pronounced trend observed throughout our original experiments.
>
>
> 	### WGA for Exponential loss in CMNIST
>
> 	|Batch \ LR|1e-6|1e-5|1e-4|1e-3|1e-2|1e-1|
> 	|-|-|-|-|-|-|-|
> 	|8   |**12.9 ± 1.3**|**37.8 ± 3.4**|**61.0 ± 1.4**|0.0 ± 0.0|0.0 ± 0.0|0.0 ± 0.0|
> 	|16  |9.6 ± 1.3|32.8 ± 2.5|54.8 ± 3.1|**65.2 ± 0.8**|0.0 ± 0.0|0.0 ± 0.0|
> 	|32  |8.8 ± 1.3|23.8 ± 1.4|46.6 ± 3.6|64.1 ± 0.9|0.0 ± 0.0|0.0 ± 0.0|
> 	|64  |2.7 ± 0.6|15.1 ± 1.0|40.1 ± 2.8|61.7 ± 1.4|0.0 ± 0.0|0.0 ± 0.0|
> 	|128 |0.0 ± 0.0|10.1 ± 1.6|34.3 ± 2.3|57.0 ± 3.4|22.0 ± 38.2|0.0 ± 0.0|
> 	|256 |0.0 ± 0.0|9.8 ± 1.3|26.7 ± 2.1|49.5 ± 3.0|**63.3 ± 2.0**|0.0 ± 0.0|
>
>
> 	### ACC for Exponential loss in CMNIST
>
> 	|Batch \ LR|1e-6|1e-5|1e-4|1e-3|1e-2|1e-1|
> 	|-|-|-|-|-|-|-|
> 	|8   |98.0 ± 0.2|98.6 ± 0.3|99.2 ± 0.2|50.0 ± 0.0|50.0 ± 0.0|33.3 ± 27.5|
> 	|16  |98.0 ± 0.2|98.3 ± 0.3|99.1 ± 0.2|99.4 ± 0.0|50.0 ± 0.0|50.0 ± 0.0|
> 	|32  |97.9 ± 0.2|97.9 ± 0.2|99.1 ± 0.1|99.4 ± 0.0|50.0 ± 0.0|50.0 ± 0.0|
> 	|64  |91.2 ± 2.2|98.1 ± 0.2|98.7 ± 0.2|99.2 ± 0.2|50.0 ± 0.0|50.0 ± 0.0|
> 	|128 |59.1 ± 7.2|97.9 ± 0.3|98.4 ± 0.3|99.1 ± 0.2|66.5 ± 27.5|50.0 ± 0.0|
> 	|256 |40.7 ± 8.8|97.9 ± 0.4|98.1 ± 0.3|99.1 ± 0.1|99.4 ± 0.1|50.0 ± 0.0|
>
> 2) **General loss- and architecture-agnostic theory (new Section A.4)**
> We add a new section providing a theoretical justification that is agnostic to both the choice of loss function and model architecture. In this framework, we remove assumptions specific to the 4-point construction, linear models, or a particular loss. Instead, we consider a general learning setting with majority and minority subpopulations characterized by correlation rate $\rho$, allowing for arbitrary hypothesis classes (including deep networks) and general loss functions.
>
> 	The key assumptions used in these theorems are mostly commonly observed phenomena in shortcut learning:
> 	* The minimizer of the loss on the biased dataset is a spurious solution $w_{bad}$
> 	* As a consequence, $w_{bad}$ achieves lower empirical loss than an invariant solution $w_{good}$, while the latter exhibits reduced reliance on spurious features (consistent with max-margin selection behavior).
> 	* The gradient discrepancy between majority and minority samples in minibatches is larger at $w_{bad}$ than $w_{good}$
>
> 	Within this broad framework, we prove that the implicit regularization induced by SGD, particularly with sufficiently small minibatch sizes, assigns a smaller regularization value to $w_{good}$ than to $w_{bad}$. Consequently, the effective objective of SGD reduces the loss gap between these solutions and tilts the optimization toward solutions with reduced reliance on spurious features. Whereas the corresponding term under GD amplifies this gap. We show that the quantitative analysis developed for the exponential loss in Section A.3 extends qualitatively to general losses (e.g., logistic and cross-entropy) and general hypothesis classes in the asymptotic regime $n,m \rightarrow \infty$.
> 	Importantly, the mechanism applies beyond spurious-correlation scenarios: any learning problem involving subpopulations with large gradient discrepancies may benefit from this effect.
>
> [1] Schapire, R. E., & Singer, Y. (1999). Improved boosting algorithms using confidence-rated predictions. Machine learning, 37(3), 297-336.

---

> ### Author Response · Authors · 2025-11-26
>
> Thank you very much for your reply. We sincerely appreciate the opportunity to clarify our intention. Our goal is **not** to position small-batch SGD as a substitute for explicit debiasing methods such as DFR, AFR, or EValS. Instead, we aim to highlight that **small batch sizes act as a trick** that can strengthen these methods. Thank you for noting this, we are revising the paper to ensure this point is stated clearly and cannot be misinterpreted.
>
> As shown in Table 2, on Domino-CMF, explicit methods trained with large batch sizes achieve WGA values that are at or below random chance. Yet, when **the very same algorithms are trained with small batches, they gain 25–37% absolute WGA improvement**, without any additional machinery. Our study shows that combining small batch sizes induces an implicit inductive bias that naturally counters reliance on spurious features. *This implicit bias then amplifies the effectiveness of explicit debiasing techniques.*
>
> Importantly, this insight also helps reduce the need for extensive hyperparameter sweeps in methods designed to mitigate shortcuts, we encourage practitioners to simply adopt small batch sizes to benefit from this built-in robustness.
>
> ### Domino-CMF: Small vs. Large Batch Size (From Table 2 of the paper)
>
> |Batch|ERM|DFR|AFR|Evals|
> |-|-|-|-|-|
> |8  |**59.0 ± 4.3**|**72.9 ± 4.7**|**67.3 ± 5.6**|**76.7 ± 1.7**|
> |128|36.8 ± 2.0|42.7 ± 2.7|40.3 ± 0.5|51.2 ± 1.4|
>
> For single-level spurious datasets, we include CMNIST to illustrate the contrast more clearly. Here, too, batch size 1 consistently yields the strongest WGA across all methods compared to large batch size (256) and full-batch, although, as expected, the effect is less pronounced than in multi-level spurious settings (see our response to Comment 2 for details).
>
> ### CMNIST: Small vs. Large Batch Size
>
> |Batch|ERM|DFR|AFR|Evals|
> |-|-|-|-|-|
> |1|**71.3 ± 1.2**|**84.98 ± 0.3**|**72.92 ± 1.7**|**78.3 ± 1.9**|
> |256|70.3 ± 2.5|83.21 ± 0.7|71.98 ± 1.7|76.48 ± 0.9|
> |Full|65.2 ± 0.8|79.73 ± 0.9|68.11 ± 1.1|73.37 ± 6.2|
>
> Again, thank you for your constructive feedback, we hope this clarification strengthens the paper and makes our contribution more transparent and useful to the community.

---

### Official Review · Reviewer_EN6D · 2025-10-31

**Soundness:** 4
**Presentation:** 3
**Contribution:** 3
**Rating:** 8
**Confidence:** 4

**Summary:**

This paper proposes the implicit regularization of SGD as a factor in robustness against spurious correlations. The main contribution is a theoretical analysis of SGD implicit regularization in a linear setting, under the four-point data model. The batch size and learning rate are identified as key factors in the bound. Experiments are also provided which transition the insights to neural networks, and training with small batch sizes is proposed as a trick to improve robustness.

**Strengths:**

1. The paper is well written and easy to follow. The overview of the theoretical results in Sections 2/3 has an appropriate level of detail, and the appendix is nicely organized.

2. The intuitive explanation -- that SGD controls the variance of mini-batch gradients, preventing certain mini-batches from overfitting to the gradient in the direction of the optimal majority group classifier -- makes sense and is illustrated well in Section 2.

3. The dichotomy that full-batch GD increases reliance on spurious correlations while small-batch SGD mitigates it is interesting and to my knowledge novel. More generally, this paper fills a gap in the literature on the understanding of how learning rate and batch size, in conjunction with the implicit regularization of gradient descent, affect robustness to spurious correlations.

4. The experiments in Section 4 are rigorous and interesting, both in the validation of the theory and in providing takeaways for practitioners.

**Weaknesses:**

1. A related work section is missing. This is important for contextualization of this paper’s results with the literature. A few references are provided in Section 1.1 and 2.1, but results and implications are not discussed in-depth.

    a. Some papers studying theory for gradient descent in the presence of spurious correlations which may be relevant for discussion: [5, 6, 7, 8]

    b. It would also be great to include more references on small-batch or large-LR training in the vein of [2, 3]. Also, I am particularly curious whether the community has found any other robustness benefits of small batch training (as briefly discussed in Section 2.1). I am aware of at least one reference [4] which showed that small-batch training has benefits for adversarial robustness, via a flat-minima argument. Are there more?

2. The four-point data setting is a relatively simple toy setting and has been well-studied since at least [1]. However, I believe this is acceptable for this paper, as its primary contribution is a new analysis of batch size/learning rate effects of SGD, which is still interesting in the four-point data setting.

3. From a technical perspective, it is unclear whether the proof techniques are particularly novel or sophisticated, i.e., whether this paper introduces any methods that might be generalizable beyond the scope of this paper. From what I can tell, the proofs mainly utilize existing results from KKT theory and probability/concentration, with a substantial amount of careful algebra. (Note: I do not consider this criteria as necessary for acceptance at ICLR, but it would perhaps constitute the difference between an 8 and a 10).

4. A minor critique is that only vision datasets are used for the experiments. While not strictly necessary, showing the small-batch results hold on a language dataset or two (e.g., CivilComments, MultiNLI) with a Transformer architecture would be interesting.

[1] Nagarajan et al. Understanding the failure modes of out-of-distribution generalization. ICLR 2021.

[2] Keskar et al. On Large-Batch Training for Deep Learning: Generalization Gap and Sharp Minima. ICLR 2017.

[3] Goyal et al. Accurate, Large Minibatch SGD: Training ImageNet in 1 Hour. ArXiv 2017.

[4] Yao et al. Hessian-based Analysis of Large Batch Training and Robustness to Adversaries. NeurIPS 2018.

[5] Qiu et al. Complexity Matters: Feature Learning in the Presence of Spurious Correlations. ICML 2024.

[6] Yang et al. Identifying Spurious Biases Early in Training through the Lens of Simplicity Bias. AISTATS 2024.

[7] Ye et al. Freeze then Train: Towards Provable Representation Learning under Spurious Correlations and Feature Noise. AISTATS 2023.

[8] Jain et al. Bias in Motion: Theoretical Insights into the Dynamics of Bias in SGD Training. NeurIPS 2024.

**Questions:**

1. It would be nice to make clear where the $\epsilon/b$ scaling comes from in Equation 7. I assume the $1/b$ is hidden in the $f$ term.

2. See Weakness 1a/b: how should this paper’s findings be contextualized with the broader literature on a) gradient descent and spurious correlations, and b) small-batch SGD learning?

3. Minor clarity/grammatical improvements:

    a. \citep should be used on line 52, 72, 196, 293, 410, Fig 5, etc

    b. Malformed citation on line 739, 764

    c. The word “rate” is missing in line 313

---

> ### Author Response · Authors · 2025-11-21
> **Authors’ Response to Reviewer EN6D**
>
> We would like to thank you for the detailed and thoughtful review. Your comments have been extremely helpful in strengthening the paper. We address your points below, and we would be very happy to continue the discussion further.
>
> ---
> ### ❖ Reviewer Comment 1 and Question 2
> We agree that the paper would benefit from a more comprehensive related-work section, and we will expand it to include the broader literature you highlight. Regarding the specific papers:
> 1) **Feature learning evolution in Training dynamics:** Papers [5, 6, 8] provide valuable analyses of training dynamics—in particular, how and when core versus spurious features are learned. These works show that spurious features, due to their simplicity and lower complexity, are typically learned early in training, whereas core features emerge later. They further characterize conditions under which spurious features dominate model predictions [6], or situations where learning the core feature slows, stalls, or fails entirely [5, 8]. Our work contributes a complementary perspective to this line of research by analyzing how SGD hyperparameters (batch size and learning rate) directly modulate shortcut reliance through implicit regularization—an aspect not explored in these training-phase analyses.
>
>     Paper [7] focuses on when last-layer retraining can recover invariant features under spurious correlations. While this is valuable for debiasing, it does not, to our understanding, directly inform the perspective we pursue. We would appreciate clarification if we have overlooked any relevant connection.
> 2) **Batch size and Learning rates for in distribution Generalization:** Prior work on in-distribution generalization shows that small-batch SGD typically outperforms large batches. Batch-only studies [2,12] find that for any fixed learning rate there is an optimal, usually non-large batch size, and that very large batches harm generalization. Work isolating learning rate is more limited, but [9] argues that appropriately tuned learning rates together with stochastic mini-batches introduce beneficial noise absent in full-batch training.
>
>     Most research studies batch size and learning rate jointly, showing that the effective noise scale—determined by their ratio—controls solution sharpness and hence generalization [3,10,11,13,14,15]. The implicit-regularization view of [16] complements these results by analyzing how the average SGD trajectory deviates from gradient flow, offering a distinct theoretical explanation of how learning rate and batch size shape optimization dynamics.
> 3) **Effect of Batch and learning rate on other Robustness**
> We appreciate the question, particularly the connection to adversarial robustness. It is plausible that SGD’s implicit regularization also affects other subpopulation-shift settings, such as class or attribute imbalance. Empirically, smaller batches often help in imbalanced-data scenarios [17], consistent with our findings. Interestingly, the situation is reversed in the context of label-noise robustness: the literature shows that large batch sizes and small learning rates often yield better performance under label noise, exactly opposite to the hyperparameter regime that benefits shortcut mitigation [18]. From the lens of implicit regularization, this difference is intuitive: under label noise, the “minority” samples are incorrectly labeled examples, and we do not want SGD to amplify their influence. In this case, reducing SGD’s variance-penalizing regularization (via large batch size and small learning rate) is beneficial, because it prevents the optimizer from fitting noisy, uninformative gradients.
>
>     Taken together, these results suggest that SGD’s implicit regularization interacts differently with each robustness setting, depending on which subpopulation should or should not influence the optimizer.
> ---
> ### ❖ Reviewer Comment 4
> We agree that including experiments on language datasets would further strengthen the paper. At the time of submission, we focused on spurious-correlation datasets, whereas MultiNLI and CivilComments reflect mostly attribute imbalance [1]. Below we report the preliminary findings, which are consistent with the trends observed in our vision experiments. While we were unable to complete the full set of exhaustive learning-rate sweeps, we are continuing these runs and will include the complete results in the final version.
> ### MultiNLI (LR = 1e-4)
> |Batch|Acc|WGA|
> |-|-|-|
> |8|82.18 ± 0.04|**76.75 ± 0.48**|
> |16|82.40 ± 0.07|76.58 ± 2.04|
> |32|81.74 ± 0.20|76.50 ± 0.57|
> |64|81.93 ± 0.13|75.80 ± 1.85|
> |128|80.96 ± 0.05|75.78 ± 0.68|
> |256|79.34 ± 0.02|75.17 ± 0.63|
>
> ### Civil Comments (LR = 1e-5)
> |Batch|Acc|WGA|
> |-|-|-|
> |8|91.76 ± 0.26|**60.72 ± 3.30**|
> |16|92.12 ± 0.12|59.73 ± 1.68|
> |32|92.34 ± 0.17|54.89 ± 4.39|
> |64|92.30 ± 0.02|53.40 ± 0.34|
> |128|92.02 ± 0.02|53.70 ± 0.74|
> ---
> ### ❖ Reviewer Question 1 and 3
> We apologize for these mistakes and will revise the paper accordingly to address them.

---

> > ### Author Response · Authors · 2025-11-21
> > **References**
> >
> > ### References
> >
> > [1] Yang et al. Change is hard: A closer look at subpopulation shift. ICML 2023
> >
> > [2] Keskar et al. On large-batch training for deep learning: Generalization gap and sharp minima. ICLR 2017
> >
> > [3] Goyal et al. Accurate, large minibatch SGD: Training ImageNet in 1 hour. arXiv 2017
> >
> > [5] Qiu et al. Complexity Matters: Feature Learning in the Presence of Spurious Correlations. ICML 2024.
> >
> > [6] Yang et al. Identifying Spurious Biases Early in Training through the Lens of Simplicity Bias. AISTATS 2024.
> >
> > [7] Ye et al. Freeze then Train: Towards Provable Representation Learning under Spurious Correlations and Feature Noise. AISTATS 2023.
> >
> > [8] Jain et al. Bias in Motion: Theoretical Insights into the Dynamics of Bias in SGD Training. NeurIPS 2024.
> >
> > [9] LeCun et al. Efficient backprop. Neural networks: Tricks of the trade 1998
> >
> > [10] Jastrzębski et al. Three factors influencing minima in SGD. ICLR Workshop 2018
> >
> > [11] Jastrzębski et al. Width of minima reached by stochastic gradient descent is influenced by learning rate to batch size ratio. ICANN 2018
> >
> > [12] Smith & Le. A Bayesian perspective on generalization and stochastic gradient descent. ICLR 2018
> >
> > [13] Chaudhari & Soatto. Stochastic gradient descent performs variational inference, converges to limit cycles for deep networks. ICLR 2018
> >
> > [14] Park et al. The effect of network width on stochastic gradient descent and generalization: An empirical study. ICML 2019
> >
> > [15] Marek et al. Small batch size training for language models: When vanilla SGD works, and why gradient accumulation is wasteful. NeurIPS 2025
> >
> > [16] Smith et al. On the origin of implicit regularization in stochastic gradient descent. ICLR 2021
> >
> > [17] Shwartz-Ziv et al. Simplifying neural network training under class imbalance. NeurIPS 2023
> >
> > [18] Rolnick et al. Deep learning is robust to massive label noise. arXiv 2018

---

> > > ### Comment · Reviewer_EN6D · 2025-11-21
> > >
> > > I appreciate the author's response and expect the paper will be improved with the inclusion of the detailed related work section. With respect to the author's comments on attribute imbalance [1], I would posit that the taxonomy and definitions of [1] are somewhat idiosyncratic and need not restrict the range of reasonable benchmark datasets. I appreciate the extra experiments, and it's heartening to see the small-batch results continue to hold in the language domain with (I assume) Transformer architectures.
> > >
> > > Overall, I believe this is an interesting and impactful paper which will be valuable to the ICLR community. I recommend acceptance.
> > >
> > > [1] Yang et al. Change is hard: A closer look at subpopulation shift. ICML 2023.

---

> > > > ### Author Response · Authors · 2025-11-26
> > > >
> > > > Thank you very much for your encouraging assessment. We truly appreciate your recommendation for acceptance and your thoughtful feedback. We are currently revising the paper to incorporate a more complete related-work section, as you suggested. We will also include the Multi-NLI, and CivilComments results obtained so far; it was reassuring for us as well to see that the small-batch phenomenon extends to the language domain. We believe this consistency reflects the generality of implicit regularization theorem.
> > > >
> > > > We are grateful for your supportive comments and are committed to improving the manuscript accordingly.

---

> > ### Author Response · Authors · 2025-12-03
> >
> > ### ❖ Reviewer Comment 2 and 3
> > We add a new section providing a theoretical justification that is agnostic to both the choice of loss function and model architecture. In this framework, we remove assumptions specific to the 4-point construction, linear models, or a particular loss. Instead, we consider a general learning setting with majority and minority subpopulations characterized by correlation rate $\rho$, allowing for arbitrary hypothesis classes (including deep networks) and general loss functions.
> >
> > Please refer to section A.4 General Results for more details.

---

### Official Review · Reviewer_XpZs · 2025-10-31

**Soundness:** 2
**Presentation:** 3
**Contribution:** 2
**Rating:** 4
**Confidence:** 4

**Summary:**

This paper investigates how the implicit regularization effect of stochastic gradient descent contributes to group robustness. Starting from a simple four-point linear model containing one invariant and one spurious feature, the authors theoretically show that SGD implicitly minimizes the variance of mini-batch gradients, which discourages the model from relying on spurious or shortcut features. Through analytical comparison with full-batch gradient descent, they demonstrate that SGD systematically assigns lower weights to spurious dimensions when the learning rate is moderately large. The paper further presents empirical results on multiple deep learning benchmarks (e.g., CMNIST, CelebA, Waterbirds, CIFAR10, Domino)

**Strengths:**

Within the four-point linear model, the analysis is mathematically valid and well-grounded in prior implicit-regularization theory.

The paper is clearly written and visually well-organized.

**Weaknesses:**

The entire formal analysis is restricted to a two-dimensional linear model with exponential loss, where spurious and invariant features are explicitly separable. The main results (Theorems 3.1–3.3) therefore have no direct generalization to nonlinear networks.

The paper claims that “the phenomenon extends to deep neural networks,” yet the deep network experiments are purely phenomenological and do not demonstrate the mechanism at play.

The derivation equates “mini-batch gradient variance” with “dependence on spurious features,” which is only true under the toy model’s assumptions. In higher-dimensional or nonlinear cases, gradient variance can stem from many other sources (noise, imbalance, stochasticity).

The implicit-regularization analysis assumes infinitesimal step size and small learning rate, yet the empirical improvements occur at large learning rates, outside the theoretical validity region.

The findings largely confirm existing empirical wisdom (“small batch, large lr improves robustness”) without offering new algorithmic insights or a quantifiable predictive model for hyperparameter selection.

**Questions:**

See weaknesses

---

> ### Author Response · Authors · 2025-11-21
> **Authors’ Response to Reviewer XpZs**
>
> We would like to thank you for your thoughtful review and constructive feedback. We appreciate your comments, which we believe could help us further improve the clarity and quality of the paper. We address each comment below and would be happy to clarify any remaining concerns.
>
> ---
>
> ### ❖ Reviewer Comment 1
>
> We acknowledge that our formal results are derived from a two-dimensional linear model. This choice aligns with the widely adopted four-point setting introduced in Nagarajan et al. (2021) [1], which is a standard theoretical framework for many studies in shortcut learning and OOD generalization [2,3,4]. *Crucially, the apparent simplicity of this setup is not a limitation but a strength:* it isolates invariant vs. spurious features in a controlled environment where the Bayes-optimal invariant classifier is trivial and achievable, **yet ERM still converges to a spurious-reliant max-margin solution**. This uncover the fundamental factor behind the failure of ERM and its simplicity is precisely what makes the model theoretically insightful [1]. Although several ablations in the literature consider variations such as higher-dimensional embeddings or adding Gaussian noise to data, these modifications mainly increase analytical complexity while preserving the key phenomenon: **even in a simple linearly separable setting, where the invariant feature fully determines the label and the spurious feature carries no additional information, standard ERM still relies on spurious features.**
>
> While prior work has primarily examined the influence of data-generating parameters such such as the strength of spurious correlations and the geometry of the data, **our work shifts the focus to training hyperparameters**, specifically batch size and learning rate. Our contribution is to show that these hyperparameters systematically modulate shortcut reliance, revealing a mechanism that, to the best of our knowledge, has not been analyzed even in the simplest setting.The novelty of our work was also highlighted by *Reviewer “EN6D”, and "AkY5"*.
>
> It is worth emphasizing that **the theoretical core and reasoning behind our analysis are general and not limited to linear models**. As discussed on section 2 the implicit regularization terms in equations (5) and (6), these mechanisms are not restricted to linear architectures and apply broadly to general loss functions and hypothesis classes. Using a small batch size increases the likelihood that “minority samples” (those whose spurious feature is misaligned with the label) constitute the majority in some mini-batches. In such cases, the corresponding regularization terms spike, thereby increasing the effective optimization objective. Consequently, SGD places less reliance on the spurious feature when the mini-batch size is small. Our choice to focus on linear models in the theorems is solely to provide a rigorous and transparent quantification of this qualitative reasoning and intuition, without becoming entangled in the substantially more intricate mathematics required for nonlinear models (e.g., neural networks), and to keep the results clean and readable.
>
>
> ---
>
> ### ❖ Reviewer Comment 2
>
> In this work, we revisit the recently observed empirical phenomenon that increasing the learning rate can improve group robustness, and we identify batch size as an additional, previously underexplored, contributing factor. We interpret both effects through the lens of implicit regularization, and we provide theoretical and empirical evidence for this **phenomenon that strengthening the implicit regularization induced by SGD reduces reliance on shortcut (spurious) features, thus increasing worst-group accuracy (WGA).**
>
> **To help clarify the mechanism intuitively and informal**: in stochastic updates, the implicit regularization of SGD corresponds to minimizing the average squared gradient norm across mini-batches (equation 6), which is tightly linked to gradient-variance suppression [6]. In high-spurious settings (e.g., $\rho$ = 0.05), minority examples form a very small fraction of the dataset. Under full-batch or very large-batch training, the minority gradients are dominated by majority gradients, yielding an update direction aligned primarily with spurious features. Conversely, with small mini-batches, there is non-negligible probability that a mini-batch contains a disproportionately high share of minority samples, even exceeding 50%, despite the global base rate. In such mini-batches, the update direction is aligned with invariant (core) features, effectively injecting corrective steps that counteract the shortcut-aligned gradients. Over training, this yields lower gradient variance across subsets, which is precisely what the implicit regularization term captures.
>
> *(Continued in the next comment)*

---

> > ### Author Response · Authors · 2025-11-21
> > **Authors’ Response to Reviewer XpZs (continued)**
> >
> > Importantly, our empirical results support this interpretation: batches and learning-rate settings that increase implicit regularization are the same ones that achieve systematically higher WGA *(Section 4.1, and 4.2)*. Furthermore, we directly calculate the implicit regularization term by averaging squared gradient norm across mini-batches, then use its normalized $\hat{R}$ after training and observe a negative correlation between $\hat{R}$ and WGA across configurations *(Section 4.3)*. This emperical results align with the formal results that reducing gradient variance, reflected in lower $\hat{R}$ corresponding to higher $WGA$, and therefore reduced shortcut reliance.
> >
> > We appreciate your feedback and will ensure that this intuition and interpretation is more clearly highlighted in the final version of the paper.
> >
> > ---
> >
> > ### ❖ Reviewer Comment 3
> >
> > We would like to emphasize two key points:
> >
> > 1) **Gradient variance induced by spurious features is well established and not specific to our four-point setup [9]**. Spurious features typically have weaker predictive power than the core feature but often induce a higher margin, making them easier for the model to pick up. Consequently, they produce very small gradients for many samples, while for the minority samples, those in which the spurious feature is misaligned with the label, the gradients become very large. Thus, although reliance on the spurious feature may reduce the average loss, it substantially increases the variance of gradients across mini-batches.
> >
> > 2) **Other sources of gradient variance do exist, but they affect core and spurious features symmetrically.** That is, increasing or decreasing reliance on the spurious feature does not alter those other sources. The only source of gradient variance that changes as the model shifts its reliance between the spurious and core features is the mechanism described above.
> >
> >    It is worth noting that our empirical comparison across balanced vs. group-imbalanced datasets (Figure 2 and Appendix Figure 6) provides supporting evidence: since both settings share similar levels of stochasticity and noise, the primary systematic difference is the presence of strong spurious correlation. We observe that the benefits of stronger implicit regularization (via smaller batch sizes or larger learning rates) are significantly more pronounced in group imbalanced datasets, suggesting that reducing mini-batch gradient variance is particularly more beneficial when spurious correlations are present, consistent with our interpretation.
> >
> >
> > ---
> >
> > ### ❖ Reviewer Comment 4
> >
> > We believe there may be a misunderstanding in this statement. Our implicit-regularization analysis does not assume an infinitesimal or vanishing learning rate; rather, it assumes a small but finite learning rate, $\epsilon > 0$. As shown in our theorems, the change in the spurious coefficient under GD and SGD takes the forms $A\epsilon + O(\epsilon^2)$ and $B\epsilon + O(\epsilon^2)$, respectively, where $A > 0$ and $B < 0$ (i.e., SGD reduces reliance on the spurious feature). Both terms vanish only as $\epsilon \to 0$, which is not the regime of interest. On the other hand, if $\epsilon$ becomes too large, one must explicitly account for the $O(\epsilon^2)$ terms to maintain rigorous bounds. Our analysis demonstrates that there exists a “bandwidth” for the learning rate in which GD and SGD exhibit, respectively, destructive and constructive behaviors with respect to reliance on the spurious feature, and this effect intensifies as the learning rate increases.
> >
> > ---
> >
> > ### ❖ Reviewer Comment 5
> >
> > We respectfully disagree with the characterization that our findings *“largely confirm existing empirical wisdom,”* and we appreciate the opportunity to clarify the novelty of our contributions.
> >
> > 1) **Prior empirical work has focused almost exclusively on large learning rates—not on batch size, nor on its interaction with learning rate in implicit regularization of SGD**.
> >
> > 2) **Implicit regularization has been studied only for in-distribution generalization, not for group robustness or shortcut mitigation.** Our results demonstrate that implicit regularization affects minority groups disproportionately, leading directly to gains in worst-group accuracy. An effect previously noted only in the context of in-distribution generalization may have far greater implications for shortcut mitigation. As evidenced in Table 4, the gains in worst-group accuracy ($\sim 5–10$%) consistently exceed the changes in overall accuracy ($\sim 1$%) across all datasets.
> >
> > About practical usefulness, please refer to our common response.

---

> > > ### Author Response · Authors · 2025-11-21
> > > **References**
> > >
> > > ### References
> > >
> > > [1] Nagarajan et al. Understanding the failure modes of out-of-distribution generalization. ICLR 2021
> > >
> > > [2] Puli et al. Don’t blame dataset shift! Shortcut learning due to gradients and cross entropy. NeurIPS 2023
> > >
> > > [3] Li et al. Generative classifiers avoid shortcut solutions. ICLR 2025
> > >
> > > [4] Xue et al. Understanding the robustness of multi-modal contrastive learning to distribution shift. ICLR 2024
> > >
> > > [5] Sagawa et al. An investigation of why overparameterization exacerbates spurious correlations. ICML 2020
> > >
> > > [6] Smith et al. On the origin of implicit regularization in stochastic gradient descent. ICLR 2021
> > >
> > > [7] Barrett & Dherin. Implicit gradient regularization. ICLR 2021
> > >
> > > [8] Barsbey et al. Large learning rates simultaneously achieve robustness to spurious correlations and compressibility. ICCV 2025
> > >
> > > [9] Kenfack et al. GradTune: Last-layer fine-tuning for group robustness without group annotation. Spurious Correlation & Shortcut Learning Workshop 2025

---

### Author Response · Authors · 2025-11-21
**Common Response to Reviewer Comments**

We would like to thank the reviewers again for their insightful feedback. A common concern raised by Reviewer *XpZs*, comment 5 and Reviewer *AkY5*, comment 3 relates to the **practical usefulness** of our study. We address it here.


Existing methods for mitigating shortcut learning perform well on **single-level spurious datasets** such as *CelebA and Waterbirds*. However, on more realistic datasets that contain complex or unknown spurious attributes, such as multi-level spurious correlations in Domino-CMF, these methods often perform **at or below random chance on the minority groups** [1]. This happens because, even after applying these methods, the second-level spurious attribute remains encoded in the learned representation and cannot be corrected when the algorithm only targets the first (known) spurious correlation [1].

In contrast, **small-batch ERM provides robustness implicitly**, without requiring any knowledge of which spurious attributes exist or how many levels they have. The underlying mechanism—implicit regularization—penalizes non-uniform regions of the loss landscape [2], which naturally improves performance on minority and underrepresented subpopulations that generate higher gradient variability.

As a result, in multi-level spurious settings, small batch sizes can be decisive: as shown in Table 2, they yield substantial gains of **25–37% in worst-group accuracy** on Domino-CMF across DFR, AFR, and EVaLS. This demonstrates that small-batch SGD is not merely a minor hyperparameter tweak, but **a practical and impactful robustness tool** that can be integrated in the training of the methods for when **spurious attributes are unknown, complex, or unannotatable**, conditions that arise frequently in real-world datasets.

It is important to note that multi-level and unknown spurious correlations represent a widely practical challenge, but it has very recently been highlighted in [1, 3, 4]. In many real-world datasets, we may not be aware of the specific spurious correlations present, nor can we easily identify the minority and majority subpopulations, especially when the sources of these correlations are not known. In such cases, annotating even a few samples with group labels becomes infeasible. We believe that SGD with small batch sizes is a suitable approach when spurious correlations are present but their nature remains unknown, as it helps mitigate their impact on the learning process.

---

### References

[1] Ghaznavi et al. Exploiting what trained models learn for making them robust to spurious correlations without group annotations. Spurious Correlation & Shortcut Learning Workshop 2025

[2] Prince. Understanding deep learning. MIT Press 2023

[3] Tsirigotis et al. Group robust classification without any group information. NeurIPS 2024

[4] Bayat et al. The pitfalls of memorization: When memorization hurts generalization. ICLR 2025

---

### Author Response · Authors · 2025-12-03
**We have updated the manuscript**

We sincerely thank the reviewers for their thoughtful and constructive feedback. We are especially grateful for the encouraging evaluations describing our work as novel (**EN6D, AkY5**), well written, illustrated, and organized (**EN6D, XpZs**), for recognizing the theory as well-grounded and valid (**EN6D, XpZs**), and for recommending acceptance (**EN6D, AkY5**).

We have now uploaded an updated version of the paper to OpenReview that incorporates most of the comments raised. We believe these revisions have significantly improved the manuscript. The main changes are summarized below:

1) **Generalized theory (new Section A.4)** We extend our analysis beyond the 4-point construction, linear models, and exponential loss to general losses (e.g., logistic and cross-entropy) and hypothesis classes. We show that SGD’s implicit regularization, especially with small mini-batches, assigns lower regularization to invariant solutions than to spurious-reliant ones, favoring robustness (A.4.1). We also show that GD has a reverse mechanism, and assigning smaller regularization to spurious-reliant solutions (A.4.2)


2) Softening the language in Section 5.4 to **emphasize that training with a small batch size is not a substitute for existing explicit debiasing methods, but rather serves as a better inductive bias** that could be a game changer in multi-level spurious or unknown spurious setup. We also added CMNIST results (Full-batch vs. batch 1) to the corresponding table, and reorganized the columns to better highlight the differences between small and large batch sizes across datasets and methods.


3) We add results on **MultiNLI and CivilComments** (fixed learning rate) in the main paper, with additional learning-rate settings in the appendix. The full exhaustive benchmarks at the scale of our vision experiments are ongoing and will be included in the camera-ready version.



4) We add a **Related Work** section reviewing prior work about learning rate and batch size effects on in-distribution generalization and robustness.


5) We have made several other minor clarifications and improvements. To list a few:

    * Clarifying the mechanism by which implicit regularization, specifically the reduction of gradient variance across minibatches, relates to improved group robustness in the experimental section.
    * Correcting normalization procedures in Section 5.3.
    * Fixing malformed or incorrect citations.


We thank the reviewers again for their insightful feedback, which directly guided these improvements and strengthened the final manuscript.

Best wishes,

The Authors

---

### Meta-Review · Area_Chair_Cx3x · 2026-01-02

**Summary:**

The paper studies the effect of batch size and learning rate in SGD when training on data with spurious features. The authors first propose a toy theoretical model that shows that SGD with a small batch size can learn solutions that rely less on spurious features compared to full-batch gradient descent. The authors then provide an empirical study of the impact of batch size on the worst-group accuracy in common spurious correlation datasets.

The reviews are mixed (4, 6, 8). Generally, the reviewers appreciated the core insight that the batch size can have an impact on robustness. The reviewers raised concerns about the theoretical model, which is quite toy, and disconnected from the practical experiments presented in the paper. Some reviewers also raised concerns with the novelty of the core observation that small batch and large learning rate can improve robustness. Finally, the small batch size provides a fairly small improvement over the baseline, and is much worse than dedicated methods for encouraging robustness; when robustness training is applied, it appears that small batches are no longer beneficial (Table 3).

Generally, I believe that the paper provides interesting empirical insights along with some theoretical justification. While the empirical results are not very impressive, the interplay between batch size and learning rate in the context of robustness is interesting, and I think the paper will be of interest to the ICLR community.

**Reviewer Concerns:**

Reviewers Reviewer AkY5 and EN6D both had a chance to engage with the rebuttal, and responded to the authors message. They were both positive about accepting the paper.

Reviewer XpZs didn't get a chance to respond to the authors. One of their main concerns is that the theoretical model is toy and there isn't an obvious connection to the experiments. Moreover, the reviewer believes "large lr, small batch improves robustness" is common knowledge. The authors responded with a new theoretical model in Appendix A.4 which is potentially less restrictive, but none of the reviewers had a chance to engage with that.

Overall, I do not expect that reviewer XpZs would be fully satisfied with the response, but it is likely that they wouldn't strongly oppose acceptance, given their original score, and the response from the authors.

**Reviewer Scores:**

I think it is possible that reviewer XpZs would increase their score to a 6, but it is hard to predict.

---

### Decision · Program_Chairs · 2026-01-26

Accept (Poster)